# Robustifying $\ell_\infty$ Adversarial Training to the Union of Perturbation Models

## Abstract

Classical adversarial training (AT) frameworks are designed to achieve high adversarial accuracy against a single attack type, typically $\ell_\infty$ norm-bounded perturbations. Recent extensions in AT have focused on defending against the union of multiple perturbation models but this benefit is obtained at the expense of a significant (up to $10\times$) increase in training complexity over single-attack $\ell_\infty$ AT. In this work, we expand the capabilities of widely popular single-attack $\ell_\infty$ AT frameworks to provide robustness to the union of $(\ell_\infty, \ell_2, \ell_1)$ perturbations while preserving their training efficiency. Our technique, referred to as **S**haped **N**oise **A**ugmented **P**rocessing (**SNAP**), exploits a well-established byproduct of single-attack AT frameworks – the reduction in the curvature of the decision boundary of networks. SNAP prepends a given deep net with a shaped noise augmentation layer whose distribution is learned along with network parameters using any standard single-attack AT. As a result, SNAP enhances adversarial accuracy of ResNet-18 on CIFAR-10 against the union of $(\ell_\infty, \ell_2, \ell_1)$ perturbations by $14\%$-to-$20\%$ for four state-of-the-art (SOTA) single-attack $\ell_\infty$ AT frameworks, and, for the first time, establishes a benchmark for ResNet-50 and ResNet-101 on ImageNet.

## 1 Introduction

Today *adversarial training* (AT) provides state-of-the-art (SOTA) empirical defense against adversarial perturbations. For this, adversarial perturbations are used during training to optimize a *robust* loss function [20, 41, 30, 35]. Early AT frameworks [20, 41] were $7\times$-to-$10\times$ more computationally demanding than vanilla training. More recent works [30, 35, 40] have significantly reduced the computational demands of AT via *single-step attacks* and *superconvergence*.

However, today's AT frameworks predominantly focus on a *single-attack*, *i.e.*, they seek robustness to a single perturbation, typically $\ell_\infty$-bounded [30, 35, 37, 41, 43, 40, 39, 26, 9, 34, 42, 10, 11, 14]. This results in low performance against other perturbations such as $\ell_2, \ell_1$, or the union of $(\ell_\infty, \ell_2, \ell_1)$. Indeed, as shown in Fig. 1, four state-of-the-art (SOTA) single-attack AT frameworks (*black markers*) employing only $\ell_\infty$-bounded perturbations achieve low adversarial accuracy $\mathcal{A}_{\text{adv}}^{(U)}$ of $\approx 15\%$-to-$20\%$ against the union of $(\ell_\infty, \ell_2, \ell_1)$ perturbations. Recent extensions in AT [21, 32, 18] do seek higher $\mathcal{A}_{\text{adv}}^{(U)}$ but only at the expense of $6\times$-to-$10\times$ increase in the total training time (*blue markers in Fig. 1*). The large training time of these AT frameworks has inhibited their application to large-scale datasets such as ImageNet, *e.g.*, Maini et al. [21], Tramèr & Boneh [32] show results for MNIST and CIFAR-10 only, while Laidlaw et al. [18] only additionally show $64 \times 64$ ImageNet-100 results.

The high training time for AT frameworks arises from two sources: (i) the need to employ larger networks, *e.g.*, MSD [21] with ResNet-18 achieves higher $\mathcal{A}_{\text{adv}}^{(U)}$ than PAT [18] with ResNet-50 (see Fig. 1); and (ii) the need to incorporate multiple perturbations during each attack step and a higher

overall number of attack steps, *e.g.*, 50 in MSD [21], 20 in AVG [32]. Obviously one can always reduce the number of attack steps in MSD/AVG to proportionally reduce training time. Doing so results in training time and $\mathcal{A}_{adv}^{(U)}$ to rapidly approach the training complexity and $\mathcal{A}_{adv}^{(U)}$ of standard AT frameworks, *e.g.*, a 5-step MSD and 2-step AVG is equivalent in training time and accuracy to PGD and TRADES, respectively. Notwithstanding the expensive nature of 50-step multi-attack training, today MSD [21] achieves a SOTA $\mathcal{A}_{adv}^{(U)}$ of 47% with ResNet-18 on CIFAR-10.

This poses a question: can we approach the high robustness of multiple-attack AT such as 50-step MSD against the union of $(\ell_\infty, \ell_2, \ell_1)$ perturbations while maintaining the low training time of fast single-attack AT frameworks such as FreeAdv [30] and FastAdv [35]?

In our quest to answer this question we find that noise augmentation using adequately shaped noise within standard single-attack AT frameworks employing $\ell_\infty$-bounded perturbations significantly improves robustness against the union of $(\ell_\infty, \ell_2, \ell_1)$ perturbations. The improvement appears to be a consequence of a well-established byproduct of AT frameworks – the reduction in the curvature of the decision boundary of networks trained using single-attack AT [6, 23]. We confirm this connection by quantifying the impact of single-attack AT on the geometric orientations of different perturbations.

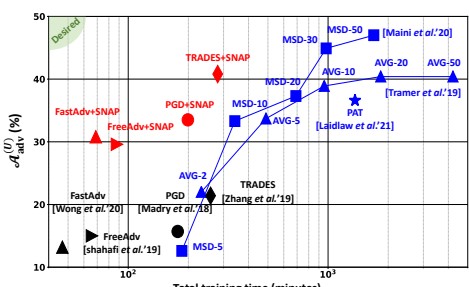

Figure 1: Adversarial accuracy $(\mathcal{A}_{adv}^{(U)})$ against union of $(\ell_\infty, \ell_2, \ell_1)$ *vs.* measured wall-clock total training time on CIFAR-10 with different AT frameworks on single NVIDIA TESLA P100 GPU. $\epsilon = (0.031, 0.5, 12)$ for $(\ell_\infty, \ell_2, \ell_1)$ perturbations, respectively. SNAP enhances robustness with a small increase in training time. All frameworks except PAT employ ResNet-18.

Based on this insight, we propose **S**haped **N**oise **A**ugmented **P**rocessing (**SNAP**) – *a method to enhance robustness against the union of perturbation types by augmenting single-attack AT frameworks.* SNAP prepends a deep net with a shaped noise (SN) augmentation layer (see Fig. 4) whose distribution parameter $\Sigma$ is learned with that of the network ($\theta$) within any standard single-attack AT framework. SNAP improves the robustness of four SOTA $\ell_\infty$-AT frameworks against the union of $(\ell_\infty, \ell_2, \ell_1)$ perturbations by 15%-to-20% on CIFAR-10 (*red markers in Fig. 1*) with only a modest ($\sim 10\%$) increase in training time. This expands the capabilities of widely popular single-attack $\ell_\infty$ AT frameworks to providing robustness to the union of $(\ell_\infty, \ell_2, \ell_1)$ perturbations without sacrificing training efficiency. We validate SNAP's benefits via thorough comparisons with *nine SOTA adversarial training and randomized smoothing frameworks* across different operating regimes on both CIFAR-10 and ImageNet.

One tangible outcome of our work – we demonstrate *for the first time* ResNet-50 (ResNet-101) networks on ImageNet that achieve $\mathcal{A}_{adv}^{(U)} = 32\%$ (35%) against the union of $(\ell_\infty(\epsilon = 2/255), \ell_2(\epsilon = 2.0), \ell_1(\epsilon = 72.0))$ perturbations. Our code and trained models will be shared publicly on GitHub.

## 2 Related Work

We categorize works on adversarial vulnerability of DNNs as follows:

**Low-complexity adversarial training**: The high computational needs of AT frameworks has spurred significant efforts in reducing their complexity [40, 30, 35, 43]. FreeAdv [30] updates weights while accumulating multiple attack iterations. FastAdv [35] employs *appropriate* use of single-step attacks, while Zheng et al. [43] leverage inter-epoch similarity between adversarial perturbations. However, these fast AT methods seek robustness against a single perturbation type, *e.g.*, $\ell_\infty$ norm-bounded perturbations. In contrast, SNAP expands the capabilities of these AT frameworks by enhancing robustness to the union of three perturbation types $(\ell_\infty, \ell_2, \ell_1)$, while preserving their efficiency.

**Robustness against union of perturbation models**: The focus on the robustness against the union of multiple perturbation types is relatively new. Kang et al. [16] studied transferability between different perturbation types, while Jordan et al. [15] considered combination attacks with low perceptual distortion. Stutz et al. [31] proposed a modification in AT to *detect* images with different models of perturbations via confidence thresholding, but they don't attempt to *classify* perturbed images correctly. For accurate classification in the presence of different perturbation models, Tramèr &

Boneh [32] studied empirical and theoretical trade-offs involved in including multiple perturbation types simultaneously during training. Maini et al. [21] further built upon this work to propose the multi steepest descent (MSD) AT framework which chooses one among the three perturbation models $(\ell_\infty, \ell_2, \ell_1)$ in each attack iteration during training, achieving SOTA adversarial accuracy on CIFAR-10 against the union of the $(\ell_\infty, \ell_2, \ell_1)$ perturbation models, albeit at a high $(10\times)$ training time. In contrast, SNAP provides high robustness against the union of $(\ell_\infty, \ell_2, \ell_1)$ perturbation models using established single-attack $\ell_\infty$ AT frameworks. This enables to showcase the benefits of our approach on large-scale datasets such as ImageNet.

Recently, Laidlaw et al. [18] developed a novel AT framework (PAT) with low perceptual distortion attacks to demonstrate impressive generalization to unseen attacks. In contrast, we focus on extending the capabilities of widely popular $\ell_\infty$-AT frameworks to providing robustness against the union of $(\ell_\infty, \ell_2, \ell_1)$ perturbations, while preserving their training efficiency.

**Noise augmentation**: Multiple recent works have investigated the role of randomization in enhancing adversarial robustness [12, 24, 8, 25] with theoretical guarantees. Another prominent line of work in this category is randomized smoothing [5, 29, 19, 38], where random noise is used as a tool to compute certification bounds. Rusak et al. [28] also explored the role of noise augmentation for improving the robustness against common-corruptions [13]. In contrast, in SNAP, noise augmentation is used as a means to enable widely popular $\ell_\infty$-AT frameworks to efficiently achieve high robustness against the union of multiple norm-bounded perturbations. As is the characteristic of AT works, our results are primarily empirical in nature. Hence, we follow recent guidelines [33, 21] to evaluate the accuracy against the strongest possible adversaries. We do explicitly compare $\ell_\infty$-AT+SNAP with randomized smoothing approaches in the Appendix.

# 3 Subspace Analysis of Adversarial Perturbations

In this section, we employ subspace methods to comprehend the distinction between $\ell_\infty$, $\ell_2$ and $\ell_1$ perturbations. For each input $x_i \in \mathbb{R}^D$ in dataset $X$, consider adversarial perturbations $\alpha_i$, $\beta_i$, and $\gamma_i$ bounded within $\ell_\infty$, $\ell_2$, and $\ell_1$ norms, respectively.

We begin with a hypothesis (see Fig. 2): *The perturbations $\alpha$, $\beta$, and $\gamma$ corresponding to input $x$ have directions that differ significantly if the curvature of the decision boundary is high in the neighborhood of $x$. Conversely, if the curvature of the decision boundary is low, the perturbations $\alpha$, $\beta$, and $\gamma$ tend to point in similar directions.*

Since, prior works [6, 23] have found that single-attack AT reduces the curvature of the decision boundary, we test our hypothesis by studying the following two networks on CIFAR-10 data: a *non-robust* ResNet18 $f_\theta^{\mathrm{van}}$ trained using vanilla training, and a *robust* ResNet18 $f_\theta^{\mathrm{rob}}$ trained using the TRADES [41] AT framework employing $\ell_\infty$ perturbations.

High curvature in decision boundary
*(vanilla training)*

Low curvature in decision boundary
*(adversarial training)*

**Perturbation models**
$\alpha$: $\ell_\infty$ norm bounded; $\beta$: $\ell_2$ norm bounded; $\gamma$: $\ell_1$ norm bounded

Figure 2: Illustration of the role of decision boundary curvature on the distinction between different types of perturbations $\alpha$, $\beta$ and $\gamma$ of the given input $x$.

We compute perturbations $\alpha_i$, $\beta_i$, and $\gamma_i$ for each $x_i \in X$ for both networks, *i.e.*, $\kappa \in \{\mathrm{van}, \mathrm{rob}\}$. We compute the singular vector basis $\mathcal{P}^\kappa$ for the set of $\ell_2$ bounded perturbations $\Delta^\kappa = \{\beta_1^\kappa, \ldots, \beta_{|X|}^\kappa\}$. The normalized mean squared projections of the three types of perturbation vectors on the singular vector basis $\mathcal{P}^\kappa$ of vanilla trained ResNet-18 ($\mathcal{P}^{\mathrm{van}}$)(Fig. 3(a)) and TRADES trained ResNet-18 ($\mathcal{P}^{\mathrm{rob}}$)(Fig. 3(b)) shows a clear contrast.

The perturbations of a vanilla trained network roll-off gradually to occupy a larger subspace as indicated in Fig. 3(a). Specifically, the projections of $\alpha$ and $\gamma$ occupy almost all 3000 directions in the basis $\mathcal{P}^{\mathrm{van}}$ since their mean squared projections are within $\sim 10\%$ of the maximum value $m_{\max}$. This shows that the dominant singular vectors of $\beta$ are not well-aligned with $\alpha$ and $\gamma$ in a vanilla trained network. With TRADES AT (Fig. 3(b)), however, all three types of perturbations are *squeezed* into a much *smaller* subspace spanning only the top 250 singular vectors in the perturbation basis $\mathcal{P}^{\mathrm{rob}}$. Outside these 250 dimensions, the mean squared projections fall to $< 10\%$ of their maximum value.

In summary, the results in Fig. 3 validate the hypothesis that single-attack AT increases the average alignment of different perturbation types due to the reduction in the decision boundary curvature. In

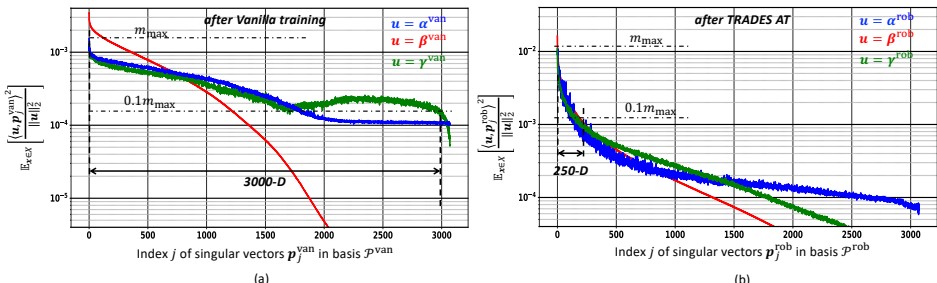

(a)

(b)

Figure 3: Normalized mean squared projections of three perturbation types on the singular vector basis $\mathcal{P}^{\kappa}$ of $\ell_2$ perturbations of ResNet18 on CIFAR-10 after: (a) vanilla training ($\kappa \equiv$ van), and (b) TRADES training ($\kappa \equiv$ rob). The singular vectors $\boldsymbol{p}_i^{\kappa}$ comprising $\mathcal{P}^{\kappa} = \{\boldsymbol{p}_1^{\kappa}, \ldots, \boldsymbol{p}_D^{\kappa}\}$ are ordered in descending order of their singular values.

144 Sec. 4, we exploit this behavior of single-attack $\ell_{\infty}$ AT to improve its robustness against the union of
145 multiple perturbation models via SNAP.

# 4 Shaped Noise Augmented Processing (SNAP)

147 We show that single-attack AT can be enhanced to address
148 multiple perturbations by introducing noise to appropriately
149 *wiggle* the $\ell_{\infty}$-bounded perturbations (Fig. 4(a)). However,
150 to do so, the noise distribution needs to be *chosen* and *shaped*
151 appropriately to minimize its impact on natural accuracy and
152 robustness to $\ell_{\infty}$-bounded perturbations.

153 We experiment with both $\ell_{\infty}$ and $\ell_2$ perturbations in single-
154 attack AT frameworks and find $\ell_{\infty}$-AT to be suitable for
155 our proposed shaped noise augmentation (see Sec. 5.2.1 for
156 details). Hence, in this section, we describe SNAP for single-
157 attack AT frameworks employing $\ell_{\infty}$ perturbations.

## 4.1 SNAPnet

159 A deep net $f_{\theta}(\boldsymbol{x}) : \mathbb{R}^D \to \{0,1\}^C$ parametrized by $\theta$ maps
160 the input $\boldsymbol{x} \in \mathbb{R}^D$ to a one-hot vector $\boldsymbol{y} \in \{0,1\}^C$ over $C$
161 classes.

162 We construct a SNAP-based deep net (SNAPnet) $f_{\theta,\Sigma}^{\text{SN}}(\boldsymbol{x})$ by
163 introducing an additive shaped noise (SN) layer (Fig. 4(b)),
164 where the noise distribution parameter $\Sigma$ is learned during
165 training. Formally,

$$\boldsymbol{y} = f_{\theta,\Sigma}^{\text{SN}}(\boldsymbol{x}) = f_{\theta}(\boldsymbol{x} + \mathbf{n}) = f_{\theta}(\boldsymbol{x} + V\Sigma\mathbf{n}_0), \quad (1)$$

166 where $\mathbf{n}_0 \sim \mathcal{L}(0, \mathbf{I}_{D \times D})$ is a zero-mean isotropic Laplace
167 noise vector, $\Sigma = \text{Diag}[\sigma_1, \ldots, \sigma_D]$ is a distribution param-
168 eter denoting its per-dimension standard deviation, $\mathbf{I}_{D \times D}$

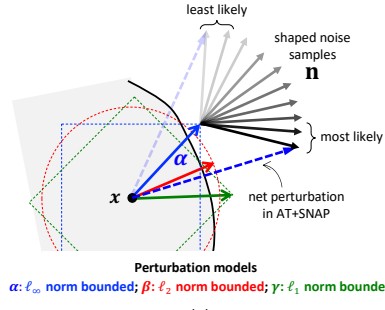

(a)

**Perturbation models**
$\boldsymbol{\alpha}$: $\ell_{\infty}$ norm bounded; $\boldsymbol{\beta}$: $\ell_2$ norm bounded; $\boldsymbol{\gamma}$: $\ell_1$ norm bounded

(b)

Figure 4: SNAP: (a) intuition underlying SNAP (not an exact depiction), and (b) SNAPnet $f_{\theta,\Sigma}^{\text{SN}}(\boldsymbol{x})$ constructed from a given deep net $f_{\theta}(\boldsymbol{x})$ by prepending a shaped noise (SN) augmentation layer which perturbs the primary input $\boldsymbol{x}$ with noise $\mathbf{n}$ whose distribution parameter $\Sigma$ is learned during AT along with the base network parameter $\theta$.

169 denotes the $D \times D$ identity matrix, and $V = [\boldsymbol{v}_1, \ldots, \boldsymbol{v}_D]$ denotes a basis in $\mathbb{R}^D$. We also studied
170 Gaussian and Uniform distributed $\mathbf{n}_0$, but empirically find the Laplace distribution to yield better
171 results (Sec. 5.2.1). We use $V = \mathbf{I}_{D \times D}$ for all our experiments in the main text and study other
172 options for $V$ in the Appendix.

173 The final classification decision $d$ is computed via

$$d = \arg\max_{c} \left[ \mathbb{E}_{\mathbf{n}} [\boldsymbol{y}] \right]_c, \quad (2)$$

174 where $[\boldsymbol{a}]_c$ denotes the $c$-th element of vector $\boldsymbol{a}$. Note, the shaped noise perturbs the input $\boldsymbol{x}$ with a
175 noise source $\mathbf{n} = V\Sigma\mathbf{n}_0$ (Eq. (1)). The distribution parameter $\Sigma$ is learned in the presence of any
176 standard AT method [20, 41, 30] used for learning deep net parameters $\theta$ as described next.

---

**Algorithm 1** Training SNAPnet

---

**Input:** training set $X$; basis $V = [\boldsymbol{v}_1, \ldots, \boldsymbol{v}_D]$; total noise power $P_{\text{noise}}$; minibatch size $r$; baseline training method BASE; noise variance update frequency $U_f$; Total number of epochs $T$

**Initialize:** noise variances $\Sigma_0 = \text{Diag}[\sigma_{1,0}, \ldots, \sigma_{D,0}]$.

**Output:** robust network $f_{\theta,\Sigma}^{\text{SN}}$, noise variances $\Sigma_T = \text{Diag}[\sigma_{1,T}^2, \ldots, \sigma_{D,T}^2]$.

1: **for** epoch $t = 1 \ldots T$ **do**

2:     **for** mini-batch $B = \{\boldsymbol{x}_1, \ldots, \boldsymbol{x}_r\}$ **do**     $\theta \leftarrow \text{BASE}_{\ell_\infty}\left(f_{\theta,\Sigma_t}^{\text{SN}}\left(\{\boldsymbol{x}_i\}_{i=1}^r\right), \theta\right)$       ▷ BASE() Training

3:     **end for**

4:     **if** $t \mod U_f = 0$ **then**                ▷ SNAP Distribution Update once every $U_f$ epochs

5:        **for** mini-batch $B = \{\boldsymbol{x}_1, \ldots, \boldsymbol{x}_r\}$ **do**

6:             $\{\boldsymbol{x}_i^{\text{adv}}\}_{i=1}^r \leftarrow \text{PGD}_{\ell_2}^{(K)}\left(f_{\theta,\Sigma_t}^{\text{SN}}\left(\{\boldsymbol{x}_i\}_{i=1}^r\right)\right);$    $\boldsymbol{\eta}_i = \boldsymbol{x}_i^{\text{adv}} - \boldsymbol{x}_i \;\; \forall \; i \in \{1, \ldots, r\}$

7:             $\gamma_j \leftarrow \gamma_j + \sum_{i=1}^r \left(\langle \boldsymbol{v}_j, \boldsymbol{\eta}_i \rangle\right)^2 \;\; \forall j \in \{1, \ldots, D\}$      ▷ Accumulate projections; See Eq. (3)

8:        **end for**

9:         $\sigma_{j,t+1}^2 = P_{\text{noise}} \frac{\sqrt{\gamma_j}}{\sum_{k=1}^D \sqrt{\gamma_k}} \;\; \forall j \in \{1, \ldots, D\}$     ▷ Normalize accumulated projections; See Eq. (3)

10:     **else**

11:         $\Sigma_{t+1} \leftarrow \Sigma_t$

12:     **end if**

13: **end for**

---

## 4.2 Training SNAPnet

Algorithm 1 summarizes the procedure for training SNAPnet $f_{\theta,\Sigma}^{\text{SN}}(\boldsymbol{x})$. In each epoch, an arbitrary AT method BASE() (line 2) updates network parameters $\theta$ with input perturbed by noise $\mathbf{n}$. Here BASE() can be any established AT framework [20, 41, 30, 35] employing $\ell_\infty$ perturbation.

The SNAP parameter $\Sigma$ is updated once every $U_f = 10$ epochs via a *SNAP distribution update* (lines 4-10). In this update, the per-dimension noise variance $\sigma_j^2$ is updated proportional to the root mean squared projection of the adversarial perturbations $\boldsymbol{\eta}$ on the basis $V$ given a total noise constraint $\sum_{j=1}^D \sigma_j^2 = P_{\text{noise}}$, where $P_{\text{noise}}$ denotes the total noise power. Formally,

$$\sigma_j^2 \propto \sqrt{\mathbb{E}_{\boldsymbol{x} \in X}\left(\langle \boldsymbol{\eta}, \boldsymbol{v}_j \rangle^2\right)} \quad \text{s.t.} \quad \sum_{j=1}^D \sigma_j^2 = P_{\text{noise}}, \tag{3}$$

where $\boldsymbol{\eta}$ is the $\ell_2$ norm-bounded PGD adversarial perturbation for the given input $\boldsymbol{x} \in X$ (line 6). Note that these $\ell_2$ perturbations are employed *only* for noise shaping and are distinct from the $\ell_\infty$ perturbations employed by BASE() AT (line 2). Also, $\ell_\infty$ perturbations cannot be used here since their projections are constant $\forall j$ when $V = \mathbf{I}_{D \times D}$, whereas employing $\ell_1$ perturbations leads to poor shaping due to high sparsity.

Thus, in SNAP, the average squared $\ell_2$ norm of the noise vector $\mathbf{n}$ is held constant at $P_{\text{noise}}$ while adapting the noise variances in the individual dimensions so as to align the noise vectors with the adversarial perturbations *on average*. Intuitively, the decision boundary is pushed aggressively in those directions.

## 4.3 Remarks

Note that the SNAP distribution update is distinct from BASE() AT. Hence, SNAP doesn't require any hyperparameter tuning in BASE(). For fairness to baselines we keep all hyperparameters identical when introducing SNAP in all our experiments. However, SNAP introduces a new hyperparameter $P_{\text{noise}}$, which permits to trade adversarial robustness $\mathcal{A}_{\text{adv}}^{(U)}$ for natural accuracy $\mathcal{A}_{\text{nat}}$. This trade-off is explored in Sec. 5.2.2.

The computational overhead of SNAP is small ($\sim 10\%$) since the *SNAP Distribution Update* occurs once in 10 epochs using just 20% of the training data to update the noise standard deviations $\sigma_j$. We provide more details about the *SNAP Distribution Update* in the Appendix.

| Method | $\mathcal{A}_{\text{nat}}$ | $\mathcal{A}_{\text{adv}}^{(\ell_\infty)}$ $\epsilon = 0.03$ | $\mathcal{A}_{\text{adv}}^{(\ell_2)}$ $\epsilon = 0.5$ | $\mathcal{A}_{\text{adv}}^{(\ell_1)}$ $\epsilon = 12$ | $\mathcal{A}_{\text{adv}}^{(U)}$ |
|---|---|---|---|---|---|
| **PGD AT with $\ell_\infty$ perturbations** | | | | | |
| PGD | 84.6 | **48.8** | 62.3 | 15.0 | 15.0 |
| +SNAP[G] | 80.7 | 45.7 | 66.9 | 34.6 | 31.9 |
| +SNAP[U] | **85.1** | 42.7 | **66.7** | 28.6 | 26.6 |
| +SNAP[L] | 83.0 | 44.8 | **68.6** | 40.1 | 35.6 |
| **PGD AT with $\ell_2$ perturbations** | | | | | |
| PGD | **89.3** | 28.8 | **67.3** | 31.8 | 25.1 |
| +SNAP[G] | 83.0 | **35.0** | 65.8 | 39.9 | 30.2 |
| +SNAP[U] | 86.4 | 32.3 | 66.7 | 30.2 | 25.0 |
| +SNAP[L] | 84.8 | 33.4 | 66.1 | **42.5** | **30.8** |

Table 1: ResNet-18 CIFAR-10 results showing the impact of SNAP augmentation of PGD [20] AT framework with $\ell_\infty$ (*top*) and $\ell_2$ (*bottom*) perturbations where [G], [U], and [L], denote shaped Gaussian, Uniform, and Laplace noise.

| Method | $\mathcal{A}_{\text{nat}}$ | $\mathcal{A}_{\text{adv}}^{(\ell_\infty)}$ $\epsilon = 0.03$ | $\mathcal{A}_{\text{adv}}^{(\ell_2)}$ $\epsilon = 0.5$ | $\mathcal{A}_{\text{adv}}^{(\ell_1)}$ $\epsilon = 12$ | $\mathcal{A}_{\text{adv}}^{(U)}$ |
|---|---|---|---|---|---|
| **High Complexity AT with $\ell_\infty$ perturbations** | | | | | |
| PGD | **84.6** | **48.8** | 62.3 | 15.0 | 15.0 |
| +SNAP | 83.0 | 44.8 | **68.6** | **40.1** | **35.6** |
| TRADES | **82.1** | **50.2** | 59.6 | 19.8 | 19.7 |
| +SNAP | 80.9 | 45.2 | **66.9** | **46.6** | **41.2** |
| **Low Complexity AT with $\ell_\infty$ perturbations** | | | | | |
| FreeAdv | 81.7 | **46.1** | 59 | 15.0 | 15.0 |
| +SNAP | **83.5** | 39.7 | **66.2** | **34.3** | **29.6** |
| FastAdv | **85.7** | **46.2** | 60.0 | 13.2 | 13.2 |
| +SNAP | 84.2 | 40.4 | **67.9** | **36.6** | **30.8** |

Table 2: ResNet-18 CIFAR-10 results showing the impact of SNAP augmentation of established $\ell_\infty$-AT frameworks. The computational overhead of SNAP is limited to $\sim 10\%$.

# 5 Experimental Results

## 5.1 Setup

Following experimental settings of prior work [41, 30, 21], we employ a ResNet-18 network for CIFAR-10 experiments and both ResNet-50 and ResNet-101 networks for ImageNet experiments. Accuracy on clean test data is referred to with $\mathcal{A}_{\text{nat}}$ and accuracy on adversarially perturbed test data is referred to via $\mathcal{A}_{\text{adv}}^{(\ell_\infty)}$, $\mathcal{A}_{\text{adv}}^{(\ell_2)}$, and $\mathcal{A}_{\text{adv}}^{(\ell_1)}$, for $\ell_\infty$, $\ell_2$, and $\ell_1$ norm bounded perturbations, respectively. Accuracy against the *union* of all three perturbations is denoted by $\mathcal{A}_{\text{adv}}^{(U)}$.

For a fair robustness comparison, our evaluation setup closely follows the setup of Maini et al. [21] for CIFAR-10 data: (1) choose norm bounds $\epsilon = (0.031, 0.5, 12.0)$ for $(\ell_\infty, \ell_2, \ell_1)$ perturbations, respectively; (2) scale norm bounds for images to lie between $[0, 1]$; (3) choose the PGD attack configuration to be *100 iterations with 10 random restarts* for all perturbation types[1]; and (4) estimate $\mathcal{A}_{\text{adv}}^{(U)}$ as the fraction of test data that is *simultaneously* resistant to all three perturbation models.

Following the guidelines of Tramer et al. [33], we carefully design *adaptive* PGD attacks that target the full defense – SN layer – since SNAPnet is end-to-end differentiable. Specifically, we backpropagate to primary input $x$ through the SN layer (see Fig. 4). Thus, the final shaped noise distribution is exposed to the adversary. We also account for the expectation $\mathbb{E}_{\mathbf{n}}[\cdot]$ in Eq. (2) by explicitly averaging deep net logits over $N_0(= 8)$ noise samples *before* computing the gradient, which eliminates any gradient obfuscation, and is known to be the strongest attack against noise augmented models [29]. In the Appendix we also show robustness stress tests and evaluate more attacks.

On CIFAR-10 data, we compare with the following seven key SOTA AT frameworks: PGD [20], TRADES [41], FreeAdv [30], FastAdv [35], AVG [32], MSD [21], PAT [18]. We also compare with two randomized smoothing frameworks [5, 29] in the Appendix. Thanks to their GitHub code releases, we first successfully reproduce their results with a ResNet-18 network in our environment. In the case of PAT [18], we evaluate and compare with their pretrained ResNet-50 model on CIFAR-10. We compare all training times on a single NVIDIA P100 GPU. On ImageNet data, we primarily compare to FreeAdv [30]. We train ResNet-50 and its SNAPnet version with FreeAdv on a Google Cloud server with four NVIDIA P100 GPUs to compare their accuracy and training times. We will release our pretrained models and code on GitHub.

## 5.2 Ablation Studies

### 5.2.1 Impact of Noise Distribution and Model of BASE() AT Perturbations

In this subsection, we first study the impact of employing $\ell_\infty$ *vs.* $\ell_2$ perturbations in BASE AT() (see line 2 in Alg. 1) on $\mathcal{A}_{\text{adv}}^{(U)}$. For each choice, we further experiment with three distributions for the SN layer in Fig. 4(b) viz. Gaussian, Uniform, and Laplace. We don't consider $\ell_1$ perturbations in

---

[1]Following Maini et al. [21], we also run all attacks on a subset of the first 1000 test examples with 10 random restarts for CIFAR-10 data.

BASE AT() since Maini et al. [21] showed that employing $\ell_1$ single-attack AT achieves very low robustness to all attacks. We choose PGD [20] AT as BASE AT() for this ablation study. For a fair comparison across the noise distributions, we fix $P_{\text{noise}} = 160$, enforcing all noise vectors to have the same average $\ell_2$ norm. For each distribution, the noise is shaped per the procedure summarized in Alg. 1.

As observed in Table 1, $\ell_\infty$-PGD AT achieves much lower $\mathcal{A}_{\text{adv}}^{(U)}$ than $\ell_2$-PGD AT, an observation also reported by Maini et al. [21]. With SNAP, however, we find that there is an interaction between the perturbation model in PGD AT and the noise distribution in SNAP. For instance, SNAP[U] enhances $\mathcal{A}_{\text{adv}}^{(U)}$ by 11% with $\ell_\infty$-PGD AT while not achieving any improvement with $\ell_2$-PGD AT. In fact, SNAP appears to be particularly suitable for $\ell_\infty$-AT, since it always improves $\mathcal{A}_{\text{adv}}^{(U)}$ by 11%-to-20.6% irrespective of the noise distribution.

Finally, of the three noise distributions, we find the Laplace distribution to be distinctly superior, achieving the highest $\mathcal{A}_{\text{adv}}^{(U)}$ (35.6% and 30.8%) due to a significant improvement in $\mathcal{A}_{\text{adv}}^{(\ell_1)}$ for both $\ell_\infty$ and $\ell_2$ PGD AT, respectively. The superiority of the Laplace distribution in achieving high $\mathcal{A}_{\text{adv}}^{(\ell_1)}$ stems from its heavier tail compared to the Gaussian and Uniform distributions with the same variance. Shaped Laplace noise generates the highest fraction of extreme values in a given noise sample. Hence, it is more effective in improving accuracy against $\ell_1$-bounded attacks, which are the strongest when perturbing few pixels by a large magnitude [21, 32]. We discuss this further in the Appendix. Henceforth, unless otherwise mentioned, we choose Laplace noise for SNAP and $\ell_\infty$ perturbations for BASE() AT as the default setting since it achieves the highest $\mathcal{A}_{\text{adv}}^{(U)}$.

### 5.2.2 Impact of $P_{\text{noise}}$

Next, we explore the impact of the SNAP hyperparameter $P_{\text{noise}}$, which constrains the average squared $\ell_2$ norm of the noise vector $\mathbf{n}$. It enables to trade between adversarial and natural accuracy.

Fig. 5 shows that, as $P_{\text{noise}}$ increases, $\mathcal{A}_{\text{adv}}^{(\ell_1)}$ improves from 31% to 47%, accompanied by a graceful (5%) drop in $\mathcal{A}_{\text{nat}}$ and a small drop of 2% in $\mathcal{A}_{\text{adv}}^{(\ell_\infty)}$ that stabilizes to $\approx 45\%$. These results show: (1) SNAP preserves the impact of $\ell_\infty$ perturbations which is not surprising since PGD AT [20] explicitly includes those, and (2) $P_{\text{noise}}$ provides an explicit knob to control the $\mathcal{A}_{\text{nat}}$ *vs.* $\mathcal{A}_{\text{adv}}$ trade-off. Henceforth, we choose $P_{\text{noise}}$ values that incur $< 1.5\%$ drop in $\mathcal{A}_{\text{nat}}$ for all SNAP+AT experiments.

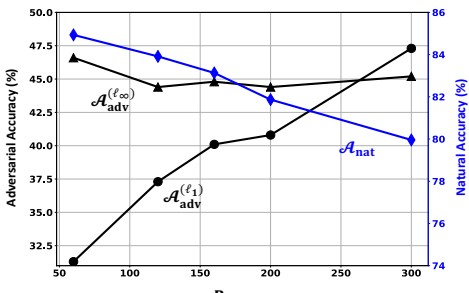

Figure 5: ResNet-18 CIFAR-10 results: adversarial accuracy $\mathcal{A}_{\text{adv}}^{(\ell_1)}$, $\mathcal{A}_{\text{adv}}^{(\ell_\infty)}$, and natural accuracy $\mathcal{A}_{\text{nat}}$ *vs.* total noise power $P_{\text{noise}}$ for PGD+SNAP.

### 5.2.3 SNAP augmented SOTA AT Frameworks

Table 2 shows the effectiveness of SNAP for four SOTA AT frameworks: high complexity frameworks, such as PGD [20], TRADES [41], and low complexity frameworks such as FreeAdv [30], FastAdv [35]. All are trained against $\ell_\infty$ attacks with $\epsilon = 0.031$. As expected, while they achieve high $\mathcal{A}_{\text{adv}}^{(\ell_\infty)}$, their $\mathcal{A}_{\text{adv}}^{(\ell_2)}$ and $\mathcal{A}_{\text{adv}}^{(\ell_1)}$ are lower.

For high-complexity AT, SNAP enhances $\mathcal{A}_{\text{adv}}^{(\ell_2)}$ and $\mathcal{A}_{\text{adv}}^{(\ell_1)}$ by $\sim 6\%$ and $\sim 25\%$, respectively, while incurring only a drop of $\sim 5\%$ in $\mathcal{A}_{\text{adv}}^{(\ell_\infty)}$. Thus overall, SNAP improves robustness ($\mathcal{A}_{\text{adv}}^{(U)}$) by $\sim 20\%$ against the *union* of the three perturbation models. Note that this robustness improvement comes at only a $\sim 1\%$ drop in $\mathcal{A}_{\text{nat}}$ (see Table 2). For low-complexity ATs, SNAP improvements in union robustness ($\mathcal{A}_{\text{adv}}^{(U)}$) are also significant ($\sim 15\%$). Again, presence of SNAP improves $\mathcal{A}_{\text{adv}}^{(\ell_2)}$ and $\mathcal{A}_{\text{adv}}^{(\ell_1)}$. This time the drop in $\mathcal{A}_{\text{adv}}^{(\ell_\infty)}$ is $\sim 7\%$. We believe this is due to the fact that these frameworks employ weaker single-step attacks during training. Note that in the case of FreeAdv+SNAP, we actually observe a $\sim 2\%$ *increase* in $\mathcal{A}_{\text{nat}}$, a trend we also observe in the ImageNet experiments described later.

| Method | LR schedule | Epochs | $\mathcal{A}_{\text{nat}}$ | $\mathcal{A}_{\text{adv}}^{(U)}$ | Total time (minutes) |
|---|---|---|---|---|---|
| **Set A: Total Time $\geq$ 12 Hrs** | | | | | |
| AVG 50 Step [32] | cyclic | 50 | 84.8 | 40.4 | 4217 |
| AVG 20 Step [32] | cyclic | 50 | **85.6** | 40.4 | 1834 |
| AVG 10 Step [32] | cyclic | 50 | **86.7** | 38.9 | **956** |
| PAT [18] | step | 100 | 82.4 | 36.6 | 1364 |
| MSD 50 Step [21] | cyclic | 50 | 81.7 | **47.0** | 1693 |
| MSD 30 Step [21] | cyclic | 50 | 82.4 | **44.9** | **978** |
| **Set B: 8 Hrs $<$ Total Time $<$ 12 Hrs** | | | | | |
| AVG 5 Step [32] | cyclic | 50 | **87.8** | 33.7 | 489 |
| MSD 20 Step [21] | cyclic | 50 | 83.0 | 37.3 | 690 |
| TRADES [41] | step | 100 | 82.0 | 19.7 | **516** |
| **TRADES+SNAP** | step | 100 | 80.9 | **41.2** | 566 |
| **Set C: 5 Hrs $<$ Total Time $<$ 8 Hrs** | | | | | |
| MSD 10 Step [21] | cyclic | 50 | 83.6 | 33.3 | **342** |
| PGD [20] | step | 100 | **84.6** | 15.0 | 354 |
| **PGD+SNAP** | step | 100 | 83.0 | **35.6** | 403 |
| **Set D: 2 Hrs $<$ Total Time $<$ 5 Hrs** | | | | | |
| AVG 2 Step [32] | cyclic | 50 | **88.4** | 22.0 | 232 |
| MSD 5 Step [21] | cyclic | 50 | **84.0** | 12.6 | **185** |
| PGD [20] | cyclic | 50 | 82.8 | 15.7 | **177** |
| TRADES [41] | cyclic | 50 | 80.0 | 21.4 | 258 |
| **PGD+SNAP** | cyclic | 50 | 82.3 | **33.5** | 199 |
| **TRADES+SNAP** | cyclic | 50 | 78.8 | **40.8** | 280 |
| **Set E: Total Time $<$ 2 Hrs** | | | | | |
| FreeAdv [30] | step | 200 | 81.7 | 15.0 | **66** |
| FastAdv [35] | cyclic | 50 | **85.7** | 13.2 | **47** |
| **FreeAdv+SNAP** | step | 200 | 83.5 | **29.6** | 88 |
| **FastAdv+SNAP** | cyclic | 50 | **84.2** | **30.8** | 69 |

Table 3: CIFAR-10 results for comparing adversarial accuracy $\mathcal{A}_{\text{adv}}^{(U)}$ *vs.* training time (on single NVIDIA P100 GPU) for different AT frameworks and the improvements by introducing proposed SNAP technique. All frameworks except PAT [18] (which employs ResNet-50) employ ResNet-18.

| Training | $\mathcal{A}_{\text{nat}}$ (%) | $\mathcal{A}_{\text{adv}}^{(\ell_\infty)}$ $\epsilon = 2/255$ | $\mathcal{A}_{\text{adv}}^{(\ell_2)}$ $\epsilon = 2.0$ | $\mathcal{A}_{\text{adv}}^{(\ell_1)}$ $\epsilon = 72.0$ | $\mathcal{A}_{\text{adv}}^{(U)}$ | Total time (minutes) |
|---|---|---|---|---|---|---|
| **ResNet-50** | | | | | | |
| FreeAdv [30] | 61.7 | **47.8** | 19.9 | 14.8 | 12.6 | **3590** |
| **FreeAdv+SNAP** | **66.8** | 46.1 | **37.8** | **37.4** | **32.4** | 3756 |
| **ResNet-101** | | | | | | |
| FreeAdv [30] | 65.4 | **51.8** | 22.8 | 18.8 | 16.1 | **5678** |
| **FreeAdv+SNAP** | **69.7** | 50.3 | **41.1** | **40.2** | **35.4** | 5904 |

Table 4: ImageNet results: Iso-hyperparameter introduction of SNAP yields $\sim 20\%$ improvement in adversarial accuracy ($\mathcal{A}_{\text{adv}}^{(U)}$) with modest impact on training time for ResNet-50 and ResNet-101.

## 5.3 Robustness *vs.* Training Complexity

Next we quantify adversarial robustness *vs.* training time trade-offs. Table 3 shows that SNAP augmentation of single-attack AT frameworks achieves the highest $\mathcal{A}_{\text{adv}}^{(U)}$, when training time is constrained to 12 hours (sets **B**, **C**, **D**, and **E**).

For instance, TRADES+SNAP achieves a 4% higher $\mathcal{A}_{\text{adv}}^{(U)}(= 41\%)$ than MSD-20 with 2 hours *lower* training time (Set **B** in Table 3). Similarly, PGD+SNAP achieves a 2% higher $\mathcal{A}_{\text{adv}}^{(U)}$ than MSD-10 while having a similar training time (Set **C**). Note that both PGD and TRADES here use 100 training epochs with standard step learning rate (LR) schedule, while MSD frameworks employ a cyclic learning rate schedule to achieve superconvergence in 50 epochs.

In Set **D**, *following* Maini et al. [21], we employ a cyclic learning rate schedule for PGD, TRADES, as well as for PGD+SNAP and TRADES+SNAP to achieve convergence in 50 epochs. Improvements in $\mathcal{A}_{\text{adv}}^{(U)}$ for PGD+SNAP and TRADES+SNAP are similar to those in Sets **B** and **C**. Most notably,

PGD+SNAP with cyclic learning rate achieves $\sim 20\%$ and $11.5\%$ *higher* $\mathcal{A}_{\text{adv}}^{(U)}$ than MSD-5 and AVG-2, respectively, while having a similar training time ($\sim 3$ hours). Set **E** augments the data from Table 2 with training times. FastAdv+SNAP and FreeAdv+SNAP achieve a high $\mathcal{A}_{\text{adv}}^{(U)} \sim 30\%$, while preserving the training efficiency of both FastAdv and FreeAdv. Notably, FastAdv+SNAP achieves $18\%$ higher $\mathcal{A}_{\text{adv}}^{(U)}$ than MSD-5, while being $\sim 2.7\times$ more efficient to train.

## 5.4 ImageNet Results

Thanks to SNAP's low computational overhead combined with FreeAdv's fast training time, we are for the first time able to report adversarial accuracy of ResNet-50 and ResNet-101 against the union of $(\ell_\infty, \ell_2, \ell_1)$ attacks on ImageNet.

We closely follow the evaluation setup of Shafahi et al. [30]. Specifically, we use 100 step PGD attack, one of the strongest adversaries considered by Shafahi et al. [30], and evaluate on the entire test set. We first reproduce FreeAdv [30] results using the *same* hyperparameters and then introduce SNAP. All hyperparameter details are specified in the Appendix.

In order to clearly demonstrate the contrast between robustness to different perturbation models, we evaluate with $\epsilon = (2/255, 2.0, 72.0)$ for $(\ell_\infty, \ell_2, \ell_1)$ attacks, respectively.[2] As shown in Table 4, FreeAdv achieves a high $\mathcal{A}_{\text{adv}}^{(\ell_\infty)} = 47.8\%$ with ResNet-50, but a lower $\mathcal{A}_{\text{adv}}^{(\ell_2)} = 20\%$ and $\mathcal{A}_{\text{adv}}^{(\ell_1)} = 15\%$, and consequently, a low $\mathcal{A}_{\text{adv}}^{(U)}$ of $12.6\%$ against the union of the perturbations. In contrast, FreeAdv+SNAP improves $\mathcal{A}_{\text{adv}}^{(\ell_2)}$ and $\mathcal{A}_{\text{adv}}^{(\ell_1)}$ by $17\%$ and $22\%$, respectively, accompanied by a $5\%$ improvement in $\mathcal{A}_{\text{nat}}$ and a small $2\%$ loss in $\mathcal{A}_{\text{adv}}^{(\ell_\infty)}$. This results in an overall robustness improvement of $20\%$ against the union of the perturbation models, setting a first benchmark for ResNet-50 on ImageNet. Upon increasing the network to ResNet-101, both natural and adversarial accuracies improve by $\approx 4\%$ for FreeAdv, a trend also observed by Shafahi et al. [30]. SNAP further improves FreeAdv's results for $\mathcal{A}_{\text{nat}}$ and $\mathcal{A}_{\text{adv}}^{(U)}$ by $4.3\%$ and $19.3\%$.

# 6 Discussion

Given the wide popularity of $\ell_\infty$-AT, in this paper, we propose SNAP as an augmentation that generalizes the effectiveness of $\ell_\infty$-AT to the union of $(\ell_\infty, \ell_2, \ell_1)$ perturbations. SNAP's strength is its simplicity and efficiency. Consequently, this work sets a first benchmark for ResNet-50 and ResNet-101 networks which are resilient to the union of $(\ell_\infty, \ell_2, \ell_1)$ perturbations on ImageNet. Note that norm-bounded perturbations include a large class of attacks, *e.g.*, gradient-based [20, 27, 32, 21, 4, 22], decision-based [3] and black-box [1] attacks.

More work is needed to extend the proposed SNAP technique to attacks beyond norm-bounded additive perturbations, *e.g.*, functional [17, 36], rotation [7], texture [2], etc. We provide preliminary evaluations in this direction in the Appendix. It is important to note that SNAP is meant to be an efficient technique for improving $\ell_\infty$-AT, and *not* a new defense. Indeed defending against a large variety of attacks simultaneously remains an open problem, with encouraging results from recent efforts [21, 18].

Another limitation of our approach is that its benefits are demonstrated empirically. It is an inevitable consequence of a lack of any theoretical guarantees for underlying AT frameworks. An interesting direction of future work is to explore whether any theoretical guarantees can be derived for anisotropic shaped noise distributions in SNAP by building upon the recent developments in randomized smoothing [29, 38]. This could be a potential avenue for bridging the gap between certification bounds and empirical adversarial accuracy.

Finally, we believe that any effort on improving adversarial robustness of deep nets has net positive societal impact. However, recent past in this field has shown that any improvements in defense techniques also lead to more effective threat models. While such a cat-and-mouse game is of great intellectual value in the academic setting, it does have an unintentional negative societal consequence of equipping malicious outside actors with a broad set of tools. This further underscores the well-recognized need for provable defenses.

---

[2]Note that $\ell_2$ and $\ell_1$ norms of PGD perturbation with $\ell_\infty$ norm of $2/255$ can be as large as $\sim 3.0$ and $\sim 1100$ for images of size $224 \times 224 \times 3$.

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
