# Supplementary Material for "Robustifying $\ell_\infty$ Adversarial Training to the Union of Perturbation Models"

# 1 Contents

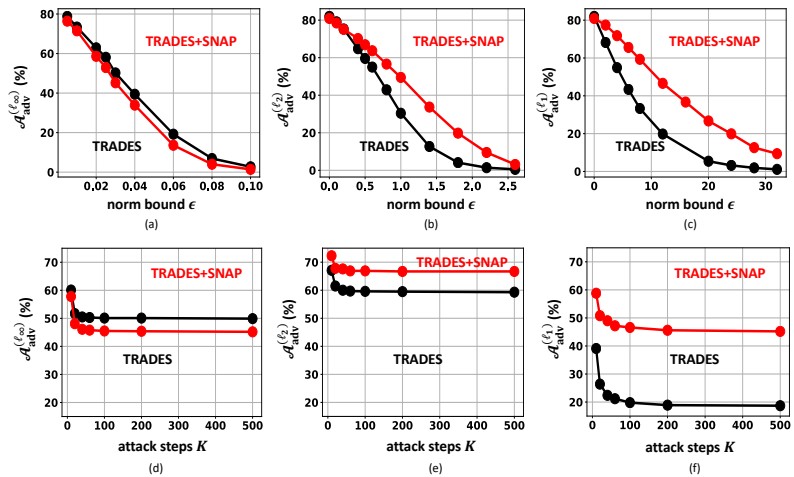

Figure I: ResNet-18 CIFAR-10 results: Adversarial accuracy *vs.* norm bound $\epsilon$ for: (a) $\ell_\infty$, (b) $\ell_2$, (c) $\ell_1$ PGD-100 attack. Adversarial accuracy *vs.* attack steps $K$ for (d) $\ell_\infty$ ($\epsilon = 0.031$), (e) $\ell_2$ ($\epsilon = 0.5$), (f) $\ell_1$ ($\epsilon = 12$) PGD-100 attacks.

## 1 Robustness Stress Tests

We conduct robustness stress tests to confirm that the benefits of SNAP are sustained for a range of attack norm-bounds, larger number of attack steps, and even for "gradient-free" attacks. For these experiments, we consider networks trained using TRADES and TRADES+SNAP (rows in Table 2 of the main paper), since they achieve the highest $\mathcal{A}_{\text{adv}}^{(U)}$ among the four SOTA AT frameworks.

### 1.1 Sweeping norm-bounds and number of attack steps

We sweep the number of PGD attack steps ($K$) and norm-bounds ($\epsilon$) for all three perturbations ($\ell_\infty, \ell_2, \ell_1$) to confirm that the robustness gains from SNAP are achieved for a wider range of attack norm bounds, and are sustained even after increasing attack steps.

Fig. I(a)-(c) validates the main text Table 2 conclusion that TRADES+SNAP achieves large gains ($\sim 20\%$) in $\mathcal{A}_{\text{adv}}^{(\ell_1)}$ and $\mathcal{A}_{\text{adv}}^{(\ell_2)}$ with a small ($\sim 4\%$) drop in $\mathcal{A}_{\text{adv}}^{(\ell_\infty)}$. Furthermore, this conclusion holds for a large range of $\epsilon$ values for all three perturbations. Additionally, the gain in $\mathcal{A}_{\text{adv}}^{(\ell_2)}$ due to SNAP at $\epsilon = 1.2$ is greater than the one reported in Table 2 for $\epsilon = 0.5$.

Now we increase the attack steps $K$ to 500 and observe the impact on adversarial accuracy against ($\ell_\infty, \ell_2, \ell_1$) perturbations in Fig. I(d,e,f), respectively. In all cases, we observe hardly any change of the adversarial accuracy beyond $K = 100$. Hence, as noted in the main text, we have chosen $K = 100$ for all our experiments in the main text and in this supplementary.

Recall we employ 10 random restarts as recommended by Maini et al. [12] for *all* our adversarial accuracy evaluations on CIFAR-10 data.

### 1.2 Evaluating robustness against new attacks

We evaluate adversarial accuracy against the recent DDN [15], Boundary [3], and Square [1] attacks. The DDN attack was shown to be one of the SOTA gradient-based attacks, while boundary attack is one of the strongest "gradient-free" attacks. Of all the attacks considered in Maini et al. [12], PGD turns out to be the strongest for $\ell_\infty$ and $\ell_1$ perturbations. Hence, in this section, we evaluate against $\ell_2$ norm-bounded DDN, boundary, and Square attacks.

Following Maini et al. [12], we use the FoolBox [13] implementation of the boundary attack, which uses 25 trials per iteration. For the DDN attack, we use 100 attack steps with appropriate logit averaging for $N_0 = 8$ noise samples *before* computing the gradient in each step (similar to our PGD attack implementations). As mentioned in the main text, it eliminates any gradient obfuscation due to the presence of noise.

|  | TRADES | **TRADES+SNAP** |
|---|---|---|
| Natural Accuracy | **82.1** | 80.9 |
| DDN [15] ($\epsilon = 0.5$) | 59.7 | **65.8** |
| Boundary [3] ($\epsilon = 0.5$) | 63.5 | **67.0** |
| Square [1] ($\epsilon = 0.5$) | 68.2 | **72.7** |

Table I: ResNet-18 CIFAR-10 results showing natural accuracy (%) and adversarial accuracy (%) against $\ell_2$ norm bounded DDN attack [15], boundary attack [3], and Square [1] for TRADES and TRADES+SNAP networks from Table 2 in the main text.

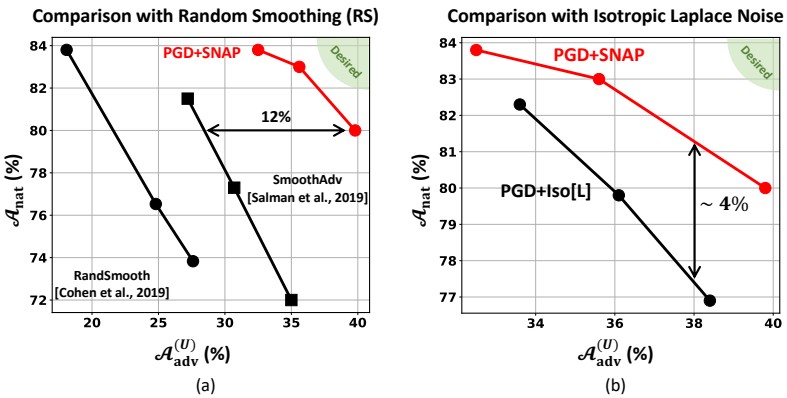

(a)  (b)

Figure II: ResNet-18 CIFAR-10 results: (a) $\mathcal{A}_{\text{nat}}$ *vs.* $\mathcal{A}_{\text{adv}}^{(U)}$ for RandSmooth [4], SmoothAdv [16], and PGD+SNAP; (b) $\mathcal{A}_{\text{nat}}$ *vs.* $\mathcal{A}_{\text{adv}}^{(U)}$ for PGD+SNAP and PGD+Iso[L], where Iso[L] denotes a baseline SNAP alternative employing isotropic Laplace noise augmentation, *i.e.*, without noise shaping. PGD+SNAP achieves better $\mathcal{A}_{\text{nat}}$ *vs.* $\mathcal{A}_{\text{adv}}^{(U)}$ trade-off due to noise shaping.

Table I shows that SNAP improves adversarial accuracy against the DDN attack by $\sim 6\%$. This is similar to improvements seen against $\ell_2$-PGD attack in Table 2 in the main text. Similarly, TRADES+SNAP achieves $3.5\%$ ($4.5\%$) higher adversarial accuracy than TRADES against the Boundary [3] (Square [1]) attack.

## 2 Additional Results

### 2.1 Comparison with Randomized Smoothing (RS)

In this subsection, we compare with two SOTA randomized smoothing (RS) works, namely, RandSmooth [4], and SmoothAdv [16]. They employ isotropic Gaussian noise. In Fig. II(a), we find that PGD+SNAP achieves a better $\mathcal{A}_{\text{nat}}$ *vs.* $\mathcal{A}_{\text{adv}}^{(U)}$ trade-off compared to both RandSmooth [4], and SmoothAdv [16]. Specifically, note that SmoothAdv [16] can also be viewed as isotropic Gaussian augmentation of $\ell_2$-PGD AT. Importantly, PGD+SNAP achieves a 12% higher $\mathcal{A}_{\text{adv}}^{(U)}$ for the same $\mathcal{A}_{\text{nat}}$. This demonstrates the efficacy of *shaped noise* in SNAP, which enhances the robustness to the union of $(\ell_\infty, \ell_2, \ell_1)$ perturbations.

In order to further quantify importance of *noise shaping*, we also compare $\ell_\infty$-PGD+SNAP with $\ell_\infty$-PGD+Iso[L], a stronger baseline alternative consisting of *isotropic* Laplace noise augmentation, *i.e.*, *without any noise shaping*. Specifically, in Iso[L], the noise standard deviation is *identical* in each direction, *i.e.*, $\Sigma = \text{Diag}\left[\sqrt{\frac{P_{\text{noise}}}{D}}, \ldots, \sqrt{\frac{P_{\text{noise}}}{D}}\right]$. Note that such distributions have recently been explored for RS [21].

Fig. II(b) plots the $\mathcal{A}_{\text{nat}}$ *vs.* $\mathcal{A}_{\text{adv}}^{(U)}$ trade-off for PGD+SNAP (*red curve*) and PGD+Iso[L] (*black curve*) by sweeping $P_{\text{noise}}$. We find that PGD+SNAP achieves a better $\mathcal{A}_{\text{nat}}$ *vs.* $\mathcal{A}_{\text{adv}}^{(U)}$ trade-off compared to PGD+Iso[L] by making more efficient use of noise power via noise shaping. Specifically, for $\mathcal{A}_{\text{adv}}^{(U)} \approx 38$, PGD+SNAP achieves a $\sim 4\%$ higher $\mathcal{A}_{\text{nat}}$.

| Method | $\mathcal{A}_{\text{nat}}$ | $\mathcal{A}_{\text{adv}}^{(\ell_\infty)}$ $\epsilon = 0.03$ | $\mathcal{A}_{\text{adv}}^{(\ell_2)}$ $\epsilon = 0.31$ | $\mathcal{A}_{\text{adv}}^{(\ell_1)}$ $\epsilon = 8$ | Time per Epoch (seconds) |
|---|---|---|---|---|---|
| MNG [10] | 79.8 | 43.9 | **75.8** | 53.8 | $354^\dagger$ |
| **PGD+SNAP** | **83.1** | **45.9** | 74.1 | **58.3** | **240** |

Table II: ResNet-18 CIFAR-10 results showing a comparison between MNG [10] and PGD+SNAP (from Table 2 in the main text). All MNG numbers are exactly as reported in their paper. We reevaluate PGD+SNAP with our PGD attacks using the new $\epsilon$ values used by Madaan et al. [10]. PGD+SNAP achieves 3%, 2%, 4.5% higher $\mathcal{A}_{\text{nat}}$, $\mathcal{A}_{\text{adv}}^{(\ell_\infty)}$, $\mathcal{A}_{\text{adv}}^{(\ell_1)}$, respectively, while being at least $\sim 40\%$ faster in terms of epoch time. †: Note that MNG time is measured on NVIDIA GeForce RTX 2080Ti (by Madaan et al. [10]), while PGD+SNAP is measured on NVIDIA Tesla P100. An RTX 2080Ti has *20% more* CUDA cores than a Tesla P100.

| Method | $\mathcal{A}_{\text{nat}}$ | $\mathcal{A}_{\text{adv}}^{(\ell_\infty)}$ $\epsilon = 0.03$ | $\mathcal{A}_{\text{adv}}^{(\ell_2)}$ $\epsilon = 0.5$ | $\mathcal{A}_{\text{adv}}^{(\ell_1)}$ $\epsilon = 8$ | $\mathcal{A}_{\text{adv}}^{(U)}$ |
|---|---|---|---|---|---|
| PGD | **89.9** | **45.3** | 34.9 | 4.8 | 4.8 |
| **PGD+SNAP** | 89.3 | 44.0 | **67.4** | **48.3** | **36.3** |

Table III: ResNet-18 SVHN results showing the impact of SNAP augmentation of $\ell_\infty$-PGD [11] AT frameworks. Adding SNAP improves $\mathcal{A}_{\text{adv}}^{(U)}$ by $\sim 30\%$ while having only a small impact on $\mathcal{A}_{\text{nat}}$ and $\mathcal{A}_{\text{adv}}^{(\ell_\infty)}$.

## 2.2 Comparison with Madaan et al. [10]

The meta-noise generator (MNG) [10] employs a multi-layer deep-net to generate noise samples during AT. Importantly, MNG still employs multiple attacks during training, but samples only one of the attacks randomly at a time to reduce the training cost.

However, they have yet to release their code or pretrained models even though their work was posted on arXiv a year ago. Absence of public codes from Madaan et al. [10] makes it difficult to clearly compare with their work, especially in terms of training time. Nonetheless, in this subsection, we try our best to ensure that the comparison is fair. Table II reports natural and adversarial accuracy of MNG against $(\ell_\infty, \ell_2, \ell_1)$ attacks as reported by Madaan et al. [10]. We find that PGD+SNAP achieves 3%, 2%, 4.5% higher $\mathcal{A}_{\text{nat}}$, $\mathcal{A}_{\text{adv}}^{(\ell_\infty)}$, and $\mathcal{A}_{\text{adv}}^{(\ell_1)}$, respectively. Note that Madaan et al. [10] evaluate $\mathcal{A}_{\text{adv}}^{(\ell_\infty)}$ and $\mathcal{A}_{\text{adv}}^{(\ell_2)}$ against PGD-50 attacks, whereas here we employ PGD-100 attacks and, following their protocol, evaluate on the entire CIFAR-10 dataset with a single restart. Furthermore, epoch time for PGD+SNAP is $1.4\times$ smaller than that of MNG [10] even though MNG time was measured on a more recent NVIDIA RTX 2080Ti, which has 20% more CUDA cores than the Tesla P100 GPU that we used for PGD+SNAP.

Importantly, a key advantage of SNAP is its scalability. We are able to report robust ResNet-50 and ResNet-101 networks on ImageNet (Table 4 in the main text), whereas Madaan et al. [10] report results only up to $64 \times 64$ TinyImageNet.

## 2.3 SVHN results

Table III shows PGD and PGD+SNAP results on SVHN data. We train both PGD and PGD+SNAP models for 100 epochs using a piece-wise LR schedule. We start with an initial LR of 0.01 and decay it once at the 95th epoch.

In Table III, we observe a trend that is similar to our observations for CIFAR-10 and ImageNet results. In particular, for SVHN, SNAP turns out to be even more effective, with $\sim 30\%$ improvement in $\mathcal{A}_{\text{adv}}^{(U)}$ while almost preserving both $\mathcal{A}_{\text{nat}}$ and $\mathcal{A}_{\text{adv}}^{(\ell_\infty)}$.

| Method | $\mathcal{A}_{\mathrm{nat}}$ (%) |
|---|---|
| TRADES | 81.7 |
| **TRADES+SNAP** | |
| $N_0 = 1$ | 80.1±0.22 |
| $N_0 = 2$ | 80.3±0.14 |
| $N_0 = 4$ | 80.7±0.12 |
| $N_0 = 8$ | 80.9±0.10 |
| $N_0 = 16$ | 80.9±0.08 |

Table IV: ResNet-18 CIFAR-10 results showing SNAP's impact on the prediction complexity, where $N_0$ denotes the number of noise samples employed to estimate $\mathbb{E}[\cdot]$ in Eq. (2) in the main text. We find that for mere accuracy estimation, even a single forward pass ($N_0 = 1$) suffices. ±xx denotes the standard deviation over 10 independent test runs.

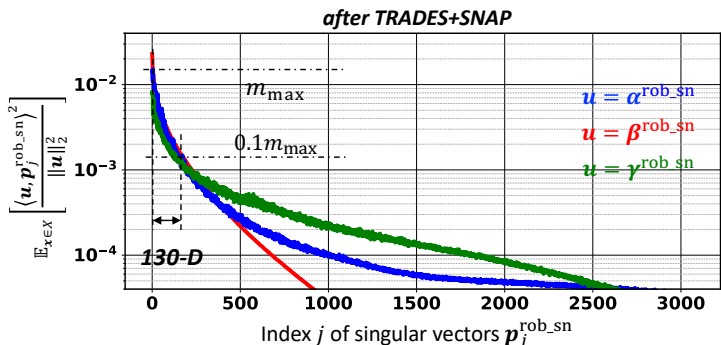

Figure III: Normalized mean squared projections of three perturbation types on the singular vector basis $\mathcal{P}^\kappa$ of $\ell_2$ perturbations of ResNet18 on CIFAR-10 after TRADES+SNAP training ($\kappa \equiv$ rob_sn). The singular vectors $\boldsymbol{p}_i^\kappa$ comprising $\mathcal{P}^\kappa = \{\boldsymbol{p}_1^\kappa, \ldots, \boldsymbol{p}_D^\kappa\}$ are ordered in descending order of their singular values.

## 2.4 Impact of SNAP on prediction complexity

While SNAP augmentation has a modest impact on the training time (Table 3 in the main text), here we check whether it could *potentially* increase the model prediction complexity due to the need to estimate the expectation $\mathbb{E}[\cdot]$ in Eq. (2) in the main text.

As expected, by increasing $N_0$, the deviation of the $\mathcal{A}_{\mathrm{nat}}$ estimate reduces (see Table IV). However, we find that for accuracy estimation, a single forward pass ($N_0 = 1$) suffices. Specifically, an $\mathcal{A}_{\mathrm{nat}}$ estimate with $N_0 = 1$ is within 1% of the $\mathcal{A}_{\mathrm{nat}}$ estimate with $N_0 = 16$. Furthermore, even with $N_0 = 1$, the standard deviation of $\mathcal{A}_{\mathrm{nat}}$ is as low as $\sim 0.2\%$. Thus, the impact of SNAP on prediction complexity can be very small.

## 2.5 Subspace analysis of adversarial perturbations for TRADES+SNAP model

In this subsection, we carry out a subspace analysis of adversarial perturbations (Section 3 in the main text) for TRADES+SNAP. We confirm that our hypothesis in Section 3 holds even after SNAP augmentation of TRADES. Following the same experimental setup and the notation from Section 3 in the main text, we compute perturbations $\boldsymbol{\alpha}_i$, $\boldsymbol{\beta}_i$, and $\boldsymbol{\gamma}_i$ for each $\boldsymbol{x}_i \in X$ for ResNet-18 trained using TRADES+SNAP, *i.e.*, $\kappa \equiv$ rob_sn. We compute the singular vector basis $\mathcal{P}^\kappa$ for the set of $\ell_2$ bounded perturbations $\Delta^\kappa = \{\boldsymbol{\beta}_1^\kappa, \ldots, \boldsymbol{\beta}_{|X|}^\kappa\}$. Fig. III plots the normalized mean squared projections of the three types of perturbation vectors on the singular vector basis $\mathcal{P}^\kappa$ of a TRADES+SNAP trained ResNet-18. We find that the projections generally follow the same trend as those for a TRADES-trained network which are shown in Fig. 3(b) of the main text. However, we also notice that after SNAP augmentation, the three perturbation types get squeezed into an even smaller 130-dimensional subspace, *i.e.*, projections are $< 10\%$ of the maximum projection value for all dimensions beyond the first 130 dimensions.

| Method | $\mathcal{A}_{\text{nat}}$ | $\mathcal{A}^{(\ell_\infty)}_{\text{adv}}$ $\epsilon = 0.03$ | $\mathcal{A}^{(\ell_2)}_{\text{adv}}$ $\epsilon = 0.5$ | $\mathcal{A}^{(\ell_1)}_{\text{adv}}$ $\epsilon = 12$ | $\mathcal{A}^{(U)}_{\text{adv}}$ |
|---|---|---|---|---|---|
| PGD | 84.6 | 48.8 | 62.3 | 15.0 | 15.0 |
| **Noise shaping basis $V = \mathbf{I}_{D \times D}$** | | | | | |
| +SNAP[G] | 80.7 | **45.7** | 66.9 | 34.6 | 31.9 |
| +SNAP[U] | **85.1** | 42.7 | **66.7** | 28.6 | 26.6 |
| +SNAP[L] | 83.0 | 44.8 | **68.6** | **40.1** | **35.6** |
| **Noise shaping basis $V = U_{\text{img}}$** | | | | | |
| +SNAP[G] | 81.7 | **48.9** | 67.5 | **29.8** | **28.7** |
| +SNAP[U] | 82.0 | 46.6 | **67.8** | 27.8 | 25.7 |
| +SNAP[L] | 81.7 | 46.8 | 65.9 | 28.5 | 27.4 |

Table V: ResNet-18 CIFAR-10 results showing the impact of noise shaping basis $V$ for $\ell_\infty$-PGD [11] AT framework with SNAP. In this table, SNAP[G], SNAP[U], and SNAP[L] denote shaped noise augmentations with Gaussian, Uniform, and Laplace noise distributions, respectively, and $U_{\text{img}}$ refers to the singular vector basis of the training images.

## 2.6 Impact of noise shaping in the image basis

Recall that, for all experiments in the main text, we chose the noise shaping basis $V = \mathbf{I}_{D \times D}$, *i.e.*, the noise was shaped and added in the standard basis in $\mathbb{R}^D$, where $\mathbf{I}_{D \times D}$ denotes the identity matrix (see Eq. (1) and Alg. 1 in the main text).

In this section, we explore the shaped noise augmentation in the *image basis*, *i.e.*, singular vector basis of the training set images. Specifically, we choose $V = U_{\text{img}} = [\boldsymbol{u}_1, \ldots, \boldsymbol{u}_D]$, where $U_{\text{img}}$ denotes the singular vector basis of the images in the training set. Thus, the sampled noise vector $\mathbf{n}_0$ (see Eq. (1) in the main text) is scaled by direction-wise standard deviation matrix $\Sigma$ and *rotated* by $U_{\text{img}}$ before being added to the input image $\boldsymbol{x}$.

The rationale for choosing $V = U_{\text{img}}$ is as follows: Recent works [7, 18, 17] have demonstrated the generative behavior of adversarial perturbations of networks trained with single-attack AT, *i.e.*, adversarial perturbations of robust networks exhibit semantics similar to the input images. Thus, the perturbation basis (see section 3 in the main text) of the robust networks trained with single-attack AT seems to be aligned with the image basis.

We repeat the experiments in Table 1 of the main text while keeping all the settings *identical* except for choosing $V = U_{\text{img}}$ instead of $V = \mathbf{I}_{D \times D}$. Table V shows the results. The first three rows correspond to $V = \mathbf{I}_{D \times D}$ and are reproduced from Table 1 of the main text. Note that, in order to preserve $\mathcal{A}_{\text{nat}} > 81\%$, we need to reduce $P_{\text{noise}} = 60$ when $V = U_{\text{img}}$, since the noise is now pixel-wise correlated.

In Table V, we notice that $\mathcal{A}^{(\ell_1)}_{\text{adv}}$ is significantly reduced when $V = U_{\text{img}}$ as compared to the case $V = \mathbf{I}_{D \times D}$. More interestingly, all three types of noise distributions result in similar values for $\mathcal{A}^{(\ell_1)}_{\text{adv}}$ when $V = U_{\text{img}}$. We discuss this phenomenon in the next section, *i.e.*, Sec. 2.7 below.

Table V shows that the orientation of a noise vector is as important as its distribution. The simpler choice of $V = \mathbf{I}_{D \times D}$ turns out to be more effective.

## 2.7 Understanding the effectiveness of SNAP[L] for $\ell_\infty$ AT

In this subsection, we conduct additional studies to further understand the following two observations in SNAP: (i) shaped Laplace noise is particularly effective (Table 1 in the main text), and (ii) rotating noise vectors ($V = U_{\text{img}}$) reduces their effectiveness (Table V in this Supplementary). We study the properties of the noise vector $\mathbf{n}$ for different noise distributions.

We conjecture that the Laplace distribution is most effective because of its heavier tail compared to Gaussian and Uniform distributions of the same variance. A long-tailed distribution will generate more large magnitude elements in a vector drawn from it and hence is more effective in emulating a

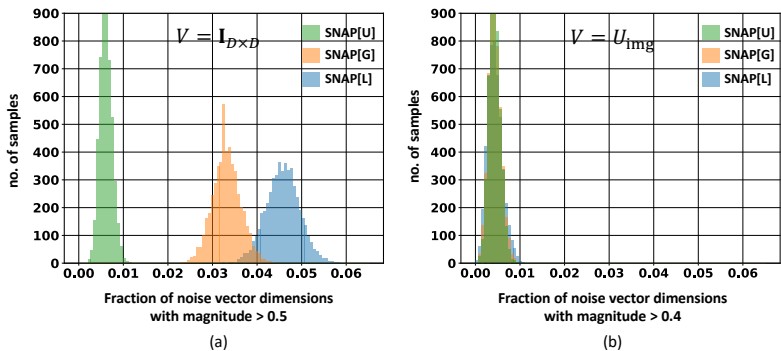

Figure IV: ResNet18 CIFAR-10 results: histograms of the fraction of noise vector dimensions with magnitude (a) $> 0.5$ when $V = \mathbf{I}_{D \times D}$, and (b) $> 0.4$ when $V = U_{\text{img}}$. Histograms are plotted for 5000 random noise samples $\mathbf{n}$. The three shaped noise distributions are from the corresponding networks in Table V.

| Method | $\mathcal{A}_{\text{nat}}$ | $\mathcal{A}_{\text{adv}}^{(U)}$ | $\mathcal{A}_{\text{cc}}$ | $\mathcal{A}_{\text{adv}}^{(f)}$ ReColorAdv |
|---|---|---|---|---|
| Vanilla | **94.5** | 0.0 | 72.0 | 0.9 |
| $\ell_\infty$-PGD | 84.6 | 15.0 | **75.6** | 53.5 |
| Noise shaping basis $V = \mathbf{I}_{D \times D}$ | | | | |
| +SNAP[G] | 80.7 | 31.9 | 72.8 | **55.1** |
| +SNAP[U] | **85.1** | 26.6 | 75.0 | 46.9 |
| +SNAP[L] | 83.0 | **35.6** | 75.3 | 51.3 |
| Noise shaping basis $V = U_{\text{img}}$ | | | | |
| +SNAP[G] | 81.7 | 28.7 | 73.6 | 54.5 |
| +SNAP[U] | 82.0 | 25.7 | 73.1 | 54.0 |
| +SNAP[L] | 81.7 | 27.4 | 73.4 | **55.3** |

Table VI: ResNet-18 CIFAR-10 results showing natural accuracy $\mathcal{A}_{\text{nat}}$, adversarial accuracy $\mathcal{A}_{\text{adv}}^{(U)}$ against the union of $(\ell_\infty, \ell_2, \ell_1)$ perturbations, accuracy $\mathcal{A}_{\text{cc}}$ in the presence of common corruptions [6], and adversarial accuracy $\mathcal{A}_{\text{adv}}^{(f)}$ against a functional adversarial attack ReColorAdv [8]. All accuracy numbers are in %. In this table, $\mathbf{I}_{D \times D}$ denotes $D$-dimensional identity matrix, while $U_{\text{img}}$ denotes singular vector basis of the training images. We find that SNAP augmentations of $\ell_\infty$-PGD significantly ($\approx 20\%$) improve $\mathcal{A}_{\text{adv}}^{(U)}$ while preserving both $\mathcal{A}_{\text{cc}}$ and $\mathcal{A}_{\text{adv}}^{(f)}$.

152 strong $\ell_1$-norm bounded perturbation. Furthermore, the standard (un-rotated) basis preserves this
153 unique attribute of samples drawn from such distributions.

154 This conjecture is validated by Fig. IV(a) which shows that noise samples drawn from the Laplace
155 distribution in the standard basis have the highest average number of dimensions with large ($> 0.5$)
156 magnitudes, followed by Gaussian and Uniform distributions. This correlates well with the results
157 in Table 1 in the main text and Table III (first three rows), in that $\mathcal{A}_{\text{adv}}^{(\ell_1)}$ is the highest for Laplace
158 followed by those for Gaussian and Uniform. Additionally, the use of $V = U_{\text{img}}$ dissolves this
159 distinction between the three distributions as shown in Fig. IV(b) which explains the similar (and
160 lower) $\mathcal{A}_{\text{adv}}^{(\ell_1)}$ values for all three distributions in Table III.

161 Thus, we confirm that the type of noise plays an important role in robustifying single-attack $\ell_\infty$ AT
162 frameworks to the union of multiple perturbation models. Specifically, the noise vectors with higher
163 fraction of noise dimensions with larger magnitudes are better at complementing $\ell_\infty$ AT frameworks.

## 2.8 Evaluating common corruptions and functional attack

165 In this subsection, we check if there are any other downsides of SNAP when it improves robustness
166 against the union of $(\ell_\infty, \ell_2, \ell_1)$ perturbations. In particular, we check if SNAP improvements are
167 achieved at the cost of a drop in accuracy against common corruptions [6] or functional adversarial
168 attacks [8].

| Method | $\mathcal{A}_{\text{nat}}$ | $\mathcal{A}_{\text{adv}}^{(\ell_\infty)}$ $\epsilon = 0.03$ | $\mathcal{A}_{\text{adv}}^{(\ell_2)}$ $\epsilon = 0.5$ | $\mathcal{A}_{\text{adv}}^{(\ell_1)}$ $\epsilon = 12$ | $\mathcal{A}_{\text{adv}}^{(U)}$ |
|---|---|---|---|---|---|
| **PGD+SNAP** | 82.5±0.27 | 43.1±0.61 | 66.9±0.57 | 39.0±0.41 | 33.7±0.29 |
| **FreeAdv+SNAP** | 83.4±0.25 | 39.2±0.74 | 65.7±0.55 | 36.5±0.60 | 30.4±0.83 |

Table VII: ResNet-18 CIFAR-10 results showing the mean and standard deviation for all accuracies over four different training runs of PGD+SNAP (with superconvergence) and FreeAdv+SNAP. As observed, the standard deviation in accuracy is $\approx 0.5\%$ in almost all cases, demonstrating the ease of replicating SNAP results.

We use corrupted images provided by Hendrycks & Dieterich [6] to estimate accuracy in the presence of common corruptions ($\mathcal{A}_{\text{cc}}$). We average the accuracy numbers across different corruption strengths and types. Also, we use the ReColorAdv setup of Laidlaw et al. [9] to estimate accuracy against functional adversarial attacks ($\mathcal{A}_{\text{adv}}^{(f)}$). We also make it *adaptive* to our defense framework via appropriate noise averaging (similar to our adaptive PGD attacks [16] discussed in the main text) to eliminate any gradient obfuscations. As observed in Table VI, SNAP augmentations of PGD AT generally preserve both $\mathcal{A}_{\text{cc}}$ and $\mathcal{A}_{\text{adv}}^{(f)}$. In particular, 20.6% improvement in $\mathcal{A}_{\text{adv}}^{(U)}$ via PGD+SNAP[L] (with $V = \mathbf{I}_{D \times D}$) is accompanied with the same $\mathcal{A}_{\text{cc}}$ and only a 2.2% lower $\mathcal{A}_{\text{adv}}^{(f)}$ ($= 51.3\%$) compared to PGD AT. In contrast, vanilla training achieves an $\mathcal{A}_{\text{adv}}^{(f)}$ of only 0.9%. Even with $V = U_{\text{img}}$, PGD+SNAP[L] achieves a 1.8% higher $\mathcal{A}_{\text{adv}}^{(f)}$ along with a 12.4% improvement in $\mathcal{A}_{\text{adv}}^{(U)}$. Note that all $\mathcal{A}_{\text{adv}}^{(U)}$ numbers are identical to the ones reported in Sec. 2.6.

We conclude that SNAP augmentation of PGD AT improves $\mathcal{A}_{\text{adv}}^{(U)}$ by up to 20% while preserving its robustness against common corruptions and functional adversarial attacks. Thus, SNAP expands the capabilities of $\ell_\infty$ AT frameworks without any significant downside. However, further work is required to improve robustness to a larger class adversarial attacks, such as rotation [5], texture [2], etc., simultaneously.

## 2.9 Error bars

In this subsection, we confirm that benefits of SNAP are not specific to any particular choice of random seed. Specifically, we run both PGD+SNAP (with superconvergence) and FreeAdv+SNAP (see Table 3 in the main text) training four times with different random seeds. Table VII shows the mean accuracy and its standard deviation for each of $\mathcal{A}_{\text{nat}}$, $\mathcal{A}_{\text{adv}}^{(\ell_\infty)}$, $\mathcal{A}_{\text{adv}}^{(\ell_2)}$, $\mathcal{A}_{\text{adv}}^{(\ell_1)}$, and $\mathcal{A}_{\text{adv}}^{(U)}$ with ResNet-18 on CIFAR-10. We find that the standard deviation of accuracy is $\approx 0.5\%$ in almost all cases. This demonstrates the ease of replicating SNAP results.

# 3 Additional Details

## 3.1 Details of Hyperparameters

### 3.1.1 Attack hyperparameters

As mentioned in the main text, we follow basic PGD attack formulations of Maini et al. [12]. We further enhance them to target the full defense – SN layer – since SNAPnet is end-to-end differentiable. Specifically, we backpropagate to the primary input $x$ through the SN layer (see Fig. 4(b) in the main text). Thus, the final shaped noise distribution is exposed to the adversary. We also account for the $\mathbb{E}_{\mathbf{n}}[\cdot]$ (see Eq. (2) in the main text) by explicitly averaging deep net logits over $N_0$ noise samples *before* computing the gradient, which eliminates any gradient obfuscation, and was shown to be the strongest attack against noise augmented models [16]. We choose $N_0 = 8$ for all our attack evaluations.

For $\ell_2$ and $\ell_\infty$ PGD attacks, we choose steps size $\alpha = 0.1\epsilon$. For $\ell_1$ PGD attacks, we choose the exact same configuration as Maini et al. [12].

### 3.1.2 Training hyperparameters

As mentioned in the main text, we introduce SNAP without changing any hyperparameters of BASE() AT. All BASE() and BASE()+SNAP training runs on CIFAR-10 employ an SGD optimizer with a fixed momentum of 0.9, batch size of 250, and weight decay of $2 \times 10^{-4}$. Also, while accounting for the $\mathbb{E}_{\mathbf{n}}[\cdot]$ (see Eq. (2) in the main text), note that $N_0 = 1$ suffices during BASE()+SNAP training. Below we provide specific details for each SOTA AT framework:

**BASE() $\equiv$ PGD [11] on CIFAR-10:**

$\ell_\infty$-PGD AT employed $\ell_\infty$-bounded PGD-$K$ attack with $\epsilon = 0.031$, step size $\alpha = 0.008$, and $K = 10$. For $\ell_2$-PGD AT, we used an $\ell_2$-bounded PGD-$K$ attack with $\epsilon = 0.5$, step size $\alpha = 0.125$ and $K = 10$. Following Rice et al. [14], we employed 100 epochs for PGD AT with step learning rate (LR) schedule, where LR was decayed from 0.1 to 0.01 at epoch 96. Following Maini et al. [12], we also employed *their* cyclic LR schedule to achieve superconvergence in 50 epochs. Following Maini et al. [12], we set weight decay to $5 \times 10^{-4}$ in PGD AT.

In PGD+SNAP, the noise variances were updated every $U_f = 10$ epochs and we use $P_{\text{noise}} = 160$ in Tables 1,2, and 3 in the main text.

**BASE() $\equiv$ TRADES [22] on CIFAR-10:**

Following Zhang et al. [22], TRADES AT employed $\ell_\infty$-bounded perturbations with $\epsilon = 0.031$, step size $\alpha = 0.007$, and attack steps $K = 10$. We set TRADES parameter $1/\lambda = 5$, which controls the weighing of its robustness regularizer. It was trained for 100 epochs with a step LR schedule, where LR was decayed to $\{0.01, 0.001, 0.0001\}$ at the epochs $\{75, 90, 100\}$, respectively. Following Maini et al. [12], we also employed *their* cyclic LR schedule to achieve superconvergence in 50 epochs, while keeping all other settings identical.

In TRADES+SNAP, the noise variances were updated every $U_f = 10$ epochs and we use $P_{\text{noise}} = 120$ in Tables 2 and 3 in the main text.

**BASE() $\equiv$ FreeAdv [18] on CIFAR-10:**

Following Shafahi et al. [18], FreeAdv AT was trained for 25 epochs, each consisting of a replay of 8. It employed $\ell_\infty$ perturbations with $\epsilon = 0.031$. The learning rate was decayed to $\{0.01, 0.001, 0.0001\}$ at epochs $\{13, 19, 23\}$, respectively.

In FreeAdv+SNAP, the noise variances were updated every $U_f = 5$ epochs, since the replay of 8 scales down the total number of epochs. Also, we use $P_{\text{noise}} = 160$ in Tables 2 and 3 in the main text.

**BASE() $\equiv$ FastAdv [20] on CIFAR-10:**

Following Wong et al. [20], FastAdv AT employed a single-step $\ell_\infty$ norm bounded FGSM attack with $\epsilon = 8/255$, step size $\alpha = 10/255$, and random noise initialization. It was trained for 50 epochs with the *same* cyclic LR schedule used by Wong et al. [20]. We used a weight decay of $5 \times 10^{-4}$.

In FastAdv+SNAP, the noise variances were updated every $U_f = 10$ epochs and we use $P_{\text{noise}} = 200$ in Tables 2 and 3 in the main text.

**BASE() $\equiv$ FreeAdv [18] on ImageNet:**

Following Shafahi et al. [18], FreeAdv AT was trained for 25 epochs, each consisting of a replay of 4. It employed $\ell_\infty$ perturbations with $\epsilon = 4/255$, *identical* to the authors' original setup. The LR was decayed by 0.1 every 8 epochs, starting with the initial LR of 0.1. We used weight decay of $1 \times 10^{-4}$.

In FreeAdv+SNAP, the noise variances were updated every $U_f = 5$ epochs, since the replay of 4 scales down the total number of epochs. Also, we use $P_{\text{noise}} = 4500$ in Table 4 in the main text, which corresponds to noise standard deviation of $\sim 0.17$ per pixel *on average*.

**MSD-$K$ [12] experiments for $K \in \{30, 20, 10, 5\}$:**

Maini et al. [12] report results for only MSD-50 in their paper. We produce MSD-$K$ results using their publicly available code. While reducing the number of steps in MSD, we *appropriately* increase the step size $\alpha$ for the attack. For MSD-50, Maini et al. [12] used $\alpha = (0.003, 0.05, 1.0)$ for $(\ell_\infty, \ell_2, \ell_1)$ perturbations, respectively. We proportionately increase the step size to $\alpha = (0.005, 0.084, 1.68)$ and $\alpha = (0.0075, 0.125, 2.5)$ for MSD-30 and MSD-20, respectively.

254 For MSD-10 and MSD-5, we choose $\alpha = (0.0075, 0.125, 2.5)$, since we found that further increasing
255 the step size $\alpha$ lead to *lower* final adversarial accuracy.

256 Other than the step-size, we do *not* make any change to the original code by Maini et al. [12].

257 **AVG-$K$ [19] experiments for $K \in \{30, 20, 10, 5\}$:**

258 For AVG-50, we use the publicly available model provided by Maini et al. [12]. We produce AVG-$K$
259 results using the Maini et al. [12] code. When reducing the number of steps, we appropriately
260 increase the step size $\alpha$ for $\ell_\infty$ and $\ell_2$ perturbations. Increasing the step size for $\ell_1$ perturbations
261 resulted in significantly lower $\mathcal{A}_{\text{adv}}^{(U)}$, and thus $\alpha$ for $\ell_1$ perturbations was kept constant while reducing
262 the number of steps. For AVG-50, Maini et al. [12] used $\alpha = (0.003, 0.05, 1.0)$ for $(\ell_\infty, \ell_2, \ell_1)$
263 perturbations, respectively. We increase the $\ell_\infty$ and $\ell_2$ step sizes to set $\alpha = (0.005, 0.084, 1.0)$ and
264 $\alpha = (0.0075, 0.125, 1.0)$ for AVG-30 and AVG-20 respectively.

265 As with MSD, we do not further increase the step size $\alpha$ for AVG-10, AVG-5, and instead choose
266 $\alpha = (0.0075, 0.125, 1.0)$. Even here, we found that increasing the step size for $\ell_1$ perturbations results
267 in lower $\mathcal{A}_{\text{adv}}^{(U)}$. For AVG-2, we increase the step size for all perturbations to $\alpha = (0.024, 0.4, 8)$.

268 **PAT [9] on CIFAR-10:**

269 For comparisons with Laidlaw et al. [9], we evaluate their publicly available self-bounded ResNet-50
270 model.

### 3.2 Details about SNAP

#### 3.2.1 Distribution Update Epoch

273 In the SNAP distribution update epoch (see Algorithm 1 in the main text), we employ $\ell_2$ norm-
274 bounded PGD attack to compute perturbation vectors $\boldsymbol{\eta}$. We use only 20% of the training data, which
275 is randomly selected during every SNAP update epoch. Recall that normalized root mean squared
276 projections of $\boldsymbol{\eta}$ dictate the updated noise variances (Eq. (3) in the main text). In the following we
277 provide more details specific to CIFAR-10 and ImageNet data:

278 CIFAR-10: we employ 10 step $\ell_2$-PGD attack with $\epsilon = 1.8$ and $N_0 = 4$.

279 ImageNet: we employ 4 step $\ell_2$-PGD attack with $\epsilon = 4.0$ and $N_0 = 1$.

280 Note that $\ell_2$ norm bound $\epsilon$ for the PGD attack here does not play any role, since $\boldsymbol{\eta}$ perturbation
281 projections are normalized.

#### 3.2.2 Noise variance initialization in SNAP

283 In SNAP, we initialize the noise variances to be uniform across all dimensions. Specifically, in
284 Algorithm 1 in the main text, $\Sigma_0 = \text{Diag}\left[\sqrt{\frac{P_{\text{noise}}}{D}}, \ldots, \sqrt{\frac{P_{\text{noise}}}{D}}\right]$ for a given value of $P_{\text{noise}}$.

## 4 Accompanying Code and Pretrained Models

286 As a part of this supplementary material, we share our code to reproduce PGD+SNAP and
287 TRADES+SNAP results on CIFAR-10 (Table 2 in the main text) and FreeAdv+SNAP results on
288 ImageNet (Table 4 in the main text). We also share corresponding pretrained models to facilitate
289 quick reproduction of our results.

290 **4.1 Code and models are available at link:** https://drive.google.com/drive/folders/
291 1yBsLkpjAEw_U2dP0P3lgO92Yah7wZsQ6?usp=sharing