# OpenReview forum: "Robustifying $\ell_\infty$  Adversarial Training to the Union of Perturbation Models"
_NeurIPS.cc/2021/Conference — NeurIPS 2021 Submitted_

### Official Review · Reviewer_dv2i · 2021-07-13

**Rating:** 5
**Confidence:** 4

**Summary:**

This paper aims to improve the robustness of networks trained using ell_infty norm-based adversarial training to the union of perturbations like ell_1, ell_2, and ell_infty. And the basic idea is to add a shaped augmentation layer that can inject noise during adversarial training.

**Limitations And Societal Impact:**

Not applicable.

**Main Review:**

Originality: This paper looks like a combination of adversarial training and randomized smoothing to me so I tends not to think it is novel enough.

Quality: The main claim of this paper is that by injecting noise during AT, the model can be robust to a union of perturbations without costing much extra time. However, from the results, the improvement looks mostly come from the ell_1 perturbation. It then makes sense why the authors found that Laplacian noise is found to be the best choice because ell_1 certified robsutness is achieved by randomized smoothing with Laplacian noise.

Clarity: The writing is clear.

Significance: Given the improvement is mostly from ell_1 robustness, the reviewer kind of doubts whether it is meaningful to consider the problem in the paper, i.e., being robust to the union of perturbations efficiently.

---
Update after rebuttal:
The results authors pointed in the paper together with the new results do address my concern on the source of improvement. Thus I would like to increase my score by 1.

**Time Spent Reviewing:**

1

---

> ### Author Response · Authors · 2021-08-10
> **Authors' response to the Reviewer's comments**
>
> We thank the reviewer for valuable feedback.
>
> 1)    **SNAP’s improvement are mostly coming from increasing the robustness to $\ell_1$ perturbations:**
>
> This is not true but it is understandable why the reviewer infers this because the CIFAR-10 Tables 1 and 2 in the main text give this impression. However, this is an artifact of the chosen $\epsilon$ value of 0.5 because we follow Maini et al.’s choices in the main text. In fact, SNAP’s improvements in $\ell_2$ robustness are higher for larger values of $\epsilon$ as can be seen in the $\epsilon$ sweep in Appendix Fig. I. Additionally, our new $\ell_2$ AutoAttack results below further supports our contention that SNAP’s gains are not mostly coming from $\ell_1$ robustness:
>
> The table below presents $\mathcal{A}^{(\ell_2)}_\text{adv}$ (%) against AutoAttack for TRADES and TRADES+SNAP
>
> |$\epsilon_{\ell_2}$| TRADES | TRADES+SNAP|
> |:---:|:---:|:---:|
> | 0.1  | 79.1  | 76.9  |
> | 0.25  | 72.0  |  74.2 |
> |  0.5 | 57.9  | 65.8  |
> | 0.75  | 43.3  | 56.5  |
> |  1.0 |  27.2  |  46.9 |
> | 1.25  |  14.7 |  35.9 |
> |  1.5 | 7.1  | 27.0  |
> | 1.75  | 3.1  |  17.7 |
> | 2.0  | 1.4  | 11.2   |
> | 2.25  |  0.6 |  5.7 |
> |  2.5 | 0.3  | 2.9  |
>
>
>
> With $\epsilon_{\ell_2}=0.5$, SNAP achieves 8% improvement, but its improvement over TRADES grows to 20% when $\epsilon_{\ell_2}=1.0$.
>
> Similarly, in the case of ImageNet (Table 4 in the main text), SNAP improves both $\mathcal{A}^{(\ell_2)}_\text{adv}$ and $\mathcal{A}^{(\ell_2)}_\text{adv}$ by a similar margin of 20%. Thus, a higher robustness against the union of attacks is achieved via improvements in the robustness against *both* $\ell_2$ and $\ell_1$ perturbations.
>
>
> 2) **the reviewer kind of doubts whether it is meaningful to consider the problem in the paper, i.e., being robust to the union of perturbations efficiently.**
>
> We are surprised by this comment since multiple works in the recent past have focused on improving the robustness against the union of perturbations (Maini et al. (2020), Tramer et al. (2019), Laidlaw et al. (2021), Madaan et al. (2021)). None of these approaches were able to scale to ImageNet due to their high computational complexity. Ours is the first work to do so due to the simplicity and effectiveness of SNAP. We hope the reviewer sees the value of our work.
>
> We hope that you will consider increasing your rating in light of our responses. We are glad to continue this conversation during the discussion period.

---

> ### Author Response · Authors · 2021-08-31
> **Query**
>
> Dear Reviewer dv2i: We appreciate your review and hope that you have found our rebuttal satisfactory. If not, please do let us know of any additional concerns you may have and we will be happy to answer those. Please note that our work has been thoroughly vetted in the past and our results can be replicated by the readers using our code that will be made available after the paper decision. You have probably noticed that our work is the first to robustify ResNet-50 and ResNet-101 on ImageNet against union of norm-bounded perturbations, thanks to its low complexity. We are glad to answer any additional questions/concerns you may have including providing additional simulations results provided it is feasible to do so in the time period available.

---

### Official Review · Reviewer_ymM2 · 2021-07-16

**Rating:** 5
**Confidence:** 4

**Summary:**

This paper produces SNAP -- that augments a noise layer to networks during standard adversarial training. On 9 different adv training methods for l_inf perturbations, the authors find that adding a noise layer helps improve robustness against the union of l1, l2, l_inf perturbations. The key hypothesis that the work is built on is that since AT reduced the curvature of the decision boundary, various attacks belonging to different perturbation types get closely aligned. Using this property, the authors attempt to modify the l_inf model boundary by adding a specifically chosen noise.

**Limitations And Societal Impact:**

Yes

**Main Review:**

### After rebuttal: Rating changed to 5

### Merits:
1. The method is a simple addition to any AT method to increases robustness
2. The training time increases is only ~10% as opposed to past works that use 3x the time.
3. Reasonable accuracy improvements over standard training methods.


### Questions and Negative points
1. *Removal of Noise Layer at test time*: The authors say that using N = 8 restarts ensures that there is no gradient masking because they use EOT. My review heavily depends on the following result -- What happens when one removes the noise layer of SNAP+AT model at test time. According to the hypothesis, SNAP should only help in training, and not inference. In that case, we should not need the noise layer at inference. If the results hold, then it empirically justifies no gradient masking, however, if they don't -- it suggests that gradients are being masked and the evaluation was not sufficient.
2. *Hypothesis Graph*: Figure 3 appears problematic to me. Why should p_j^2 not be divided while graphing the expectation? It can mislead the reader into believing in higher alignment. I would like to view the graph after this change to make any inferences because it seems rather surprising that the adversarial attacks should align so well even in non-AT setting.
3. *Do adversarial vectors really align?*: It has been shown in past work that adversaries in different perturbation regions do not subsume each other: see table 3 in https://arxiv.org/pdf/1905.11213.pdf. The hypothesis that they should align seems surprising.
4. I find it difficult to understand that even if the Hypothesis in Section 3 holds, how does it motivate adding a random noise layer to "wiggle" the l_inf boundary? It seems that the two observations are disjoint. Can you help understand this link better?
5. *Evaluation*: Since Maini et. al. was first published in 2019, the standards of adversarial training evaluation have significantly changed. I request the authors to use attacks from AutoAttack library as standardized by RobustBench for l2, linf. Additionally please use, Fast Adaptive Boundary Attack (FAB) for all of l1,l2,l_inf to perform a thorough evaluation. These attacks are faster to run, and often help identify situations where gradients are masked.
6. *What about MNIST?*: It has been found in past work that hyperparameter tuning was especially tricky in case of the MNIST dataset. Does SNAP also successfully augment noise in case of AT methods on MNIST?


**Time Spent Reviewing:**

6

---

> ### Author Response · Authors · 2021-08-10
> **Authors' response to the Reviewer's comments**
>
> We thank the reviewer for the valuable and detailed feedback.
>
> 1)    **Removal of Noise Layer at test time:**
>
> There is a major misunderstanding in this question. The presence of noise layer in SNAP improves robustness in two aspects:
> - (**A1**) during $\ell_\infty$-AT+SNAP training, the noise is shaped to predominantly lie in a smaller ~250-D subspace where all three types of perturbations have higher projections, thereby introducing appropriate wiggle in that subspace to push out the decision boundary in those directions; and
> - (**A2**) during inference, the presence of SNAP noise enables **smoothed** classification decisions (see Eq. (2)).
>
> While we primarily highlight the first aspect (**A1**) (related to our hypothesis) in the paper, both of the above contribute to robustness enhancements in SNAP. The reviewer is **missing** the role of the second aspect (**A2**) above on the robustness enhancements in SNAP. When a deep net makes a smoothed classification decision, noise can improve robustness even if an adversary knows the exact optimal direction to flip the label.  Here is a toy example that shows such a case \[[anonymous link to a figure](https://docs.google.com/document/d/1Hxrh0TqacneFj-RKCpic9lYMgTfNhw6bGiNiRb1m1DI/edit?usp=sharing)\], where the *smoothed* decision (label probability) in the presence of noise does not change even though the image is moved to the network’s decision boundary (middle part). In this example, the adversary needs to employ a larger perturbation to successfully flip the *smoothed* decision (rightmost part). Hence, we do expect some drop in the robustness when the noise is removed at test time.
>
> We incorporate the reviewer's suggestion to check the robustness in the absence of test noise. This allows us to quantitatively isolate the contributions of above two aspects in SNAP’s improvements as shown below:
>
> The Table below shows $\mathcal{A}^{(\ell_2)}_\text{adv}$ (%) for TRADES, TRADES+SNAP, and TRADES+SNAP (without noise during test) against the PGD-100 attack with 10 random restarts and N=8 gradient averaging.
>
> |$\epsilon_{\ell_2}$| TRADES | TRADES+SNAP|  TRADES+SNAP (no noise during test) |
> |:---:|:---:|:---:|:---:|
> | 0.5  | 59.6  | 66.9  | 61.9  |
> | 0.8 | 42.9  | 56.6  |  46.3 |
> | 1.0  | 30.4  |  49.5 | 37.5  |
>
> The Table below shows $\mathcal{A}^{(\ell_1)}_\text{adv} $ (%) for TRADES, TRADES+SNAP, and TRADES+SNAP (without noise during test) against the PGD-100 attack with 10 random restarts and N=8 gradient averaging.
>
> |$\epsilon_{\ell_1}$| TRADES | TRADES+SNAP|  TRADES+SNAP (no noise during test) |
> |:---:|:---:|:---:|:---:|
> | 8  | 33.3  | 59.3 | 51.4  |
> | 12 | 19.8  | 46.6  |  39.2 |
> | 16  | 10.7  |  36.7 | 28.6  |
>
> As expected, we see some drop in robust accuracies when the noise is removed at test time, which corresponds to contributions from **A2**. More importantly, in spite of this drop, SNAP’s robustness without noise during inference is significantly higher compared to TRADES only indicating the contributions from **A1**. Specifically, for $\epsilon_{\ell_1}=12$, TRADES+SNAP achieves 19% higher $A^{(\ell_1)}_\text{adv}$ compared to TRADES even after removing the noise. This shows that SNAP did change the decision boundary appropriately during training. If the reviewer’s assertion was true, TRADES+SNAP robustness without test time noise would be close to TRADES alone.
>
> This observation is consistent with Randomized Smoothing works, e.g., Salman et al. (2019) and Li et al. (2019) who also report **empirical** robust accuracies in the presence of noise during test time. For example, see Fig. 3 by Salman et al. (2019) and Fig. 3 by Li et al. (2019) where the *empirical* robust accuracy improvements can be attributed to aspect **A2** above. Our evaluation setup (based on Tramer et al. (2020)) employs stronger attacks than theirs. These *empirical* robust accuracies are well-accepted in the community today and are known to be **not** due to gradient masking.
>
>
> 2)    **Hypothesis Graph; Why should p_j^2 not be divided while graphing the expectation in Fig. 3?**
>
> The vectors $\mathbf{p}_j$  are obtained via a singular vector decomposition, hence they already have unit $\ell_2$ norm, i.e., $||\mathbf{p}_j||_2 = 1$. Hence, dividing by $||\mathbf{p}_j||^2_2$ would not change the plots in Fig. 3 at all.
>
> 3)    **Do adversarial vectors really align?: see table 3 in https://arxiv.org/pdf/1905.11213.pdf  [R1].**
>
> Table 3 in [R1] that the reviewer points to only includes the magnitude comparisons of different perturbation types. It does not compare their directions at all. For example, the first row in Table 3 quantifies the “percentage of $\ell_1$-perturbations with $\ell_\infty$-norm $\leq \epsilon_\infty$”. This condition *only* depends on the vector's *norms*. Hence, Table 3 in [R1] does **not** provide any information about the directions of $\ell_1$ and $\ell_\infty$ perturbations at all.
> In contrast, our Fig. 3 is characterizing the vector *directions* of different perturbation types on average. In fact, we are normalizing with respect to their $\ell_2$ norms (note the division by $||\mathbf{u}_j||_2^2$ in the y-axis label in Fig. 3 and note that $||\mathbf{p}_j||_2^2 = 1$).
> Thus, Table 3 in [R1] and our Fig. 3 are providing entirely complementary information. The former is conducting a norm (magnitude) comparison, while the latter is comparing only the directions.
>
>
> 4)   **How does the Hypothesis in Section 3 motivate adding a random noise layer to "wiggle" the l_inf boundary?**
>
> It is unclear what the reviewer is referring to by “wiggle the $\ell_\infty$ boundary”. The way we see it, Section 3 and our SNAP method are linked as follows:
> The hypothesis and its validation in Section 3 conveys that a few directions  are more important in terms of the robustness to multiple attacks on average compared to the remaining directions in the basis. For instance, Fig. 3b shows that after $\ell_\infty$-AT, all three types of perturbations are predominantly lying in a smaller subspace of 250-D. Hence, it makes sense to add noise during $\ell_\infty$-AT and shape that noise such that it has higher variance in that smaller subspace of 250-D. This is exactly what we are doing in SNAP. Our SNAP noise introduces appropriate wiggle in that dominant subspace of 250-D (shared by all three types of perturbations) to prevent $\ell_\infty$-AT from overfitting against $\ell_\infty$ attack alone. Thus,  $\ell_\infty$-AT+SNAP achieves high robustness against multiple attacks simultaneously.
>
>
> 5)    **Evaluation against AutoAttack:**
>
> As per the reviewer’s request, we have evaluated the SNAP models against $\ell_2$ and $\ell_\infty$ bounded AutoAttacks. Here are the results:
>
> The table below presents $\mathcal{A}^{(\ell_2)}_\text{adv}$ (%) against AutoAttack (with N=8 EOT) for TRADES and TRADES+SNAP
>
> |$\epsilon_{\ell_2}$| TRADES | TRADES+SNAP|
> |:---:|:---:|:---:|
> | 0.1  | 79.1  | 76.9  |
> | 0.25  | 72.0  |  74.2 |
> |  0.5 | 57.9  | 65.8  |
> | 0.75  | 43.3  | 56.5  |
> |  1.0 |  27.2  |  46.9 |
> | 1.25  |  14.7 |  35.9 |
> |  1.5 | 7.1  | 27.0  |
> | 1.75  | 3.1  |  17.7 |
> | 2.0  | 1.4  | 11.2   |
> | 2.25  |  0.6 |  5.7 |
> |  2.5 | 0.3  | 2.9  |
>
> Similarly, the table below presents $\mathcal{A}^{(\ell_\infty)}_\text{adv}$ (%) against AutoAttack (with N=8 EOT) for TRADES and TRADES+SNAP
>
> |$\epsilon_{\ell_\infty}$| TRADES | TRADES+SNAP|
> |:---:|:---:|:---:|
> | 0.005  | 78.4  | 75.6  |
> | 0.01  | 72.5  |  69.7 |
> |  0.02 | 61.3  |  57.4  |
> | 0.031  | 47.6  | 43.2  |
> |  0.04 |  36.9  |  31.7 |
> | 0.06 |  16.0|  12.1 |
> |  0.08 | 5.9  | 2.7  |
> | 0.1  | 2.2  |  1.1 |
>
>
> As observed in the tables above, SNAP’s improvement in robustness against $\ell_2$ perturbations is sustained even against the stronger AutoAttack. Most notably, TRADES+SNAP achieves 19% higher $\mathcal{A}^{(\ell_2)}_\text{adv}$ compared to TRADES when  $\epsilon$=1.0. Furthermore, similar to the trend in our Table 1, Table 2, and Appendix Fig. I, TRADES+SNAP achieves ~5% lower robustness compared to TRADES against $\ell_\infty$ AutoAttack. We’ll include these results in the Appendix.
>
> 6) **Evaluation against FAB:**
>
> Thank you for this suggestion. We evaluated **smoothed** deep nets such as those obtained via SNAP and SmoothAdv (Salman et al., 2019) models against FAB attacks. Interestingly we find that even after employing gradient averaging via EOT, the FAB attack is ineffective in successfully fooling such **smoothed** deep nets. In both TRADES+SNAP and SmoothAdv, we find (see Tables below) that $\ell_p$-FAB $p\in${$\infty$, 2, 1} attacks are significantly ineffective compared to $\ell_p$-PGD with the same norm budget even after appropriately averaging the gradients. Note that **smoothed** deep nets were **not** considered in the original FAB attack formulation and so we conjecture that the FAB attack needs to be re-formulation to improve its effectiveness against such deep nets. This could be an interesting direction for future work.
>
> Robust accuracy (%) against $\ell_\infty$ bounded PGD and FAB attacks with $\epsilon=0.031$:
>
> |Attack| TRADES | TRADES+SNAP|  SmoothAdv |
> |:---:|:---:|:---:|:---:|
> | $\ell_\infty$-FAB  | 48.0  | 70.1  |  59.4 |
> | $\ell_\infty$-PGD  | 50.2  |  45.2 | 32.5  |
>
> Robust accuracy (%) against $\ell_2$ bounded PGD and FAB attacks with $\epsilon=0.5$:
>
> |Attack| TRADES | TRADES+SNAP|  SmoothAdv |
> |:---:|:---:|:---:|:---:|
> | $\ell_2$-FAB  | 58.8  | 78.4  | 74.8  |
> | $\ell_2$-PGD  | 59.6  | 66.9 | 60.5  |
>
> Robust accuracy (%) against $\ell_1$ bounded PGD and FAB attacks with $\epsilon=12$:
>
> |Attack| TRADES | TRADES+SNAP|  SmoothAdv |
> |:---:|:---:|:---:|:---:|
> | $\ell_1$-FAB  | 34.7 | 71.0  | 66.8  |
> | $\ell_1$-PGD  | 19.8  |  46.6 | 43.7  |
>
>
> We hope the reviewer finds our responses satisfactory and will consider improving the overall rating. We are happy to continue this conversation during the discussion period.

---

> > ### Comment · Reviewer_ymM2 · 2021-08-30
> > **Follow Up**
> >
> > I thank the authors for a detailed rebuttal. I have modified my rating for now. I have some more questions based on their rebuttal:
> > *1. (A2)* during inference, the presence of SNAP noise enables smoothed classification decisions (see Eq. (2)).
> > Although this is not an important point in the paper, it is still interesting to note if it helps. My main issue is that randomisation should not happen during the PGD iterations. I am curious if:
> > (a) You add the randomization after PGD iterations (on the adversarial image)
> > (b) You use FAB attack against the norn-randomized model.
> >
> > What would these results be like
> >
> > *2. Do adversarial vectors really align?*
> > Based on your response, do you suggest that the adversarial vectors of all the perturbation types are majorly just re-scaled versions of each other? Since the important vectors align well?

---

> > > ### Author Response · Authors · 2021-08-31
> > > **Authors' Response**
> > >
> > > 1. (A2) during inference, the presence of SNAP noise enables smoothed classification decisions (see Eq. (2)). Although this is not an important point in the paper, it is still interesting to note if it helps. My main issue is that randomisation should not happen during the PGD iterations. I am curious if:
> > >
> > > **(a) I am curious if you add the randomization after PGD iterations (on the adversarial image), what would the results be like?**
> > >
> > > If we add randomization *after* the PGD iterations, we see much higher robust accuracies, i.e., such an attack will be weaker than the one we use in the paper. Why is this -  Since PGD iterations occur *without* noise, PGD is generating an adversarial example for the *base* deep net (without noise layer). Such an adversarial example is not always effective against the **smoothed** deep net (where the decision is made in the presence of noise). The middle section in our previous  \[[anonymous link to a figure](https://docs.google.com/document/d/1Hxrh0TqacneFj-RKCpic9lYMgTfNhw6bGiNiRb1m1DI/edit?usp=sharing)\] also provides an intuition for this case.
> > > The tables below show the experimental results.
> > >
> > > The Table below shows $\mathcal{A}^{(\ell_2)}_\text{adv}$ (%) for TRADES, TRADES+SNAP, and TRADES+SNAP (no noise during PGD but the inference is in the presence of noise) against the PGD-100 attack with 10 random restarts and N=8 gradient averaging.
> > >
> > > |$\epsilon_{\ell_2}$| TRADES | TRADES+SNAP | TRADES+SNAP (no noise during PGD; noise present in test) |
> > > |:---:|:---:|:---:|:---:|
> > > | 0.5  | 59.6  | 66.9  | 70.3  |
> > > | 0.8 | 42.9  | 56.6  |  60.8  |
> > > | 1.0  | 30.4  |  49.5 |  55.5  |
> > >
> > > The Table below shows $\mathcal{A}^{(\ell_1)}_\text{adv} $ (%) for TRADES, TRADES+SNAP, and TRADES+SNAP (no noise during PGD but the inference is in the presence of noise) against the PGD-100 attack with 10 random restarts and N=8 gradient averaging.
> > >
> > > |$\epsilon_{\ell_1}$| TRADES | TRADES+SNAP|  TRADES+SNAP (no noise during PGD; noise present in test) |
> > > |:---:|:---:|:---:|:---:|
> > > | 8  | 33.3  | 59.3 |  65.3  |
> > > | 12 | 19.8  | 46.6  | 54.2   |
> > > | 16  | 10.7  |  36.7 |  46.2  |
> > >
> > > Note that, as expected, both $\mathcal{A}^{(\ell_2)}_\text{adv}$ and $\mathcal{A}^{(\ell_1)}_\text{adv} $ are 4-9% **higher** when the noise is removed during PGD iterations. Hence, the PGD attack where the noise is removed during PGD iterations is **weaker** than our standard PGD attack formulation in the paper. Also, our PGD formulation follows Salman et al. (2019) who argue that this is the strongest PGD attack against **smoothed** deep nets.
> > >
> > > **(b) How does the FAB attack evaluate against the non-randomized models?**
> > >
> > > Results with FAB attack against non-randomized model show the same trend as our PGD attack results against non-randomized model. Kindly find the experimental results below:
> > >
> > > The Tables below show $\mathcal{A}^{(\ell_2)}_\text{adv}$ (%) for TRADES and TRADES+SNAP (no noise during attack iterations as well as inference) against the PGD-100 attack with 10 random restarts and N=8 gradient averaging.
> > >
> > > - With attack norm bound $\epsilon_{\ell_2}=0.5$
> > >
> > > | Attack | TRADES |  TRADES+SNAP (no noise during attack iterations and inference) |
> > > |:---:|:---:|:---:|
> > > | $\ell_2$-PGD  | 59.6   | 61.9  |
> > > | $\ell_2$-FAB | 57.9   |  60.8 |
> > >
> > > - With attack norm bound $\epsilon_{\ell_2}=1.0$
> > >
> > > | Attack  | TRADES |  TRADES+SNAP (no noise during attack iterations and inference) |
> > > |:---:|:---:|:---:|
> > > |  $\ell_2$-PGD  | 30.4  | 37.5  |
> > > | $\ell_2$-FAB | 22.7   |  36.6 |
> > >
> > >
> > > The Tables below show $\mathcal{A}^{(\ell_1)}_\text{adv} $ (%) for TRADES, TRADES+SNAP, and TRADES+SNAP (no noise during attack iterations and inference) against the PGD-100 attack with 10 random restarts and N=8 gradient averaging.
> > >
> > > - With attack norm bound $\epsilon_{\ell_1}=8$
> > >
> > > |$\epsilon_{\ell_1}$| TRADES |  TRADES+SNAP (no noise during attack iterations and inference) |
> > > |:---:|:---:|:---:|
> > > | $\ell_1$-PGD  | 33.3  | 51.4  |
> > > | $\ell_1$-FAB | 34.7   |  53.2 |
> > >
> > > - With attack norm bound $\epsilon_{\ell_1}=16$
> > >
> > > |$\epsilon_{\ell_1}$| TRADES |  TRADES+SNAP (no noise during attack iterations and inference) |
> > > |:---:|:---:|:---:|
> > > | $\ell_1$-PGD  | 10.7  | 28.6  |
> > > | $\ell_1$-FAB  | 10.8  |  33.9  |
> > >
> > > Even after *removing* the noise during both attack iterations *and* inference, TRADES+SNAP achieves 14% higher $\mathcal{A}^{(\ell_2)}_\text{adv}$ against $\ell_2$-FAB (with norm bound 1.0) compared to TRADES. Similarly, it achieves 23% higher $\mathcal{A}^{(\ell_1)}_\text{adv}$ against $\ell_1$-FAB (with norm bound 16) compared to TRADES. This shows that the presence of SNAP during training did change the decision boundary effectively. These trends are similar to those observed with PGD attacks.
> > >
> > > 2. **Do adversarial vectors really align? Based on your response, do you suggest that the adversarial vectors of all the perturbation types are majorly just re-scaled versions of each other? Since the important vectors align well?**
> > >
> > > No, we cannot make such a strong statement based on our analysis for the following two reasons:
> > >
> > > (i) Our analysis & observations are based on **averaged** projections over the orthogonal  basis vectors, where the averaging of projections is done over the dataset. So, there will always be some images where the three perturbation types are poorly aligned. For methods like SNAP, average increase in their alignment is sufficient.
> > >
> > > (ii) we can show that the angles between three perturbation types for most images are smaller for AT trained models compared to vanilla trained models. But the perturbations of different types are still *not* really collinear for them to be literally scaled versions of each other.

---

> > > ### Author Response · Authors · 2021-09-02
> > > **Query**
> > >
> > > We appreciate your follow up questions, which we have answered to the best of our ability. Please do let us know if you have any more queries or concerns that is preventing a recommendation for acceptance. As mentioned to the other reviewers - our work has been thoroughly vetted in the past and our results can be replicated by the readers using our code that will be made available after the paper decision. You have probably noticed that our work is the first to robustify ResNet-50 and ResNet-101 on ImageNet against union of norm-bounded perturbations, thanks to its low complexity. We are glad to answer any additional questions/concerns you may have including providing additional simulations results provided it is feasible to do so in the time period available. Thanks again for your efforts in reviewing our paper.

---

### Official Review · Reviewer_d8RZ · 2021-07-16

**Rating:** 5
**Confidence:** 4

**Summary:**

This paper expands the capabilities of widely popular single-attack $\ell_{\infty}$ adversarial training (AT) frameworks to provide robustness to the union of $(\ell_1, \ell_2, \ell_{\infty})$ perturbations while preserving the training efficiency. The proposed method, named SNAP, could reduce the curvature of the decision boundary of networks. SNAP prepends a given deep net with a shaped noise augmentation layer whose distribution is learned along with network parameters using any standard single-attack AT. SNAP enhances adversarial accuracy of ResNet-18 on CIFAR-10 against the union of $(\ell_1, \ell+2, \ell_{\infty})$ perturbations for four single-attack $\ell_{\infty}$ AT frameworks, and establishes a benchmark for ResNet-50 and ResNet-101 on ImageNet.

**Limitations And Societal Impact:**

Yes.

**Main Review:**

1. $V=[v_1,\dots,v_D]$ denotes a group of basis in $\mathbb{R}^D$. Is V the orthonormal basis?

2. The authors claim that "the average squared `$\ell_2$ norm of the noise vector $n$ is held constant at $P_noise$ while adapting the noise variances in the individual dimensions to align the noise vectors with the adversarial perturbations on average." Why can such an alignment help to defend the union attack?

3. The proposed method sacrifices the performance of the clean data to improve robustness against special attacks. However, Eq. (1) can be treated as mitigating through randomization. Unfortunately, this mechanism has been broken by [1]. Therefore, the vulnerability of SNAP should be fully considered under various kinds of adversarial methods.


[1] Obfuscated Gradients Give a False Sense of Security: Circumventing Defenses to Adversarial Examples

**Time Spent Reviewing:**

4

---

> ### Author Response · Authors · 2021-08-10
> **Authors' response to the Reviewer's comments**
>
> We thank the reviewer for valuable feedback. Here is our response:
> 1)    Is V the orthonormal basis:
>
> Yes, $V = [v_1, …, v_D]$ denotes an orthonormal basis. We will clarify it on line 169.
>
> 2)    The authors claim that "the average squared‚ $\ell_2$ norm of the noise vector $\mathbf{n}$ is held constant at $P_\text{noise}$ while adapting the noise variances in the individual dimensions to align the noise vectors with the adversarial perturbations on average." Why can such an alignment help to defend the union attack?
>
> This is shown in Fig. 3, where the three perturbation types get squeezed into a smaller 250-D subspace after $\ell_\infty$-AT. Thus, the directions in this subspace are more important to defend against the union attack than the remaining directions in the basis. Based on this observation, we introduce noise during $\ell_\infty$-AT that is shaped to primarily lie in this 250-D subspace. This is achieved by changing the noise variance to make it higher in that 250-D subspace. We do so under a total noise variance budget of $P_\text{noise}$.
>
> 3) The proposed method sacrifices the performance of the clean data to improve robustness against special attacks.
>
> True. There is no “free lunch” in adversarial robustness. All $\ell_\infty$-AT methods incur a drop in $\mathcal{A}_\text{nat}$ compared to vanilla training when they improve $\mathcal{A}^{(\ell_\infty)}_\text{adv}$ (see TRADES (Zhang et al., 2019)). This is also true for SOTA approach (Maini et al., 2020) which incurs a drop $\mathcal{A}_\text{nat}$ when improving the robustness $\mathcal{A}^{(U)}_\text{adv}$ against the union of $(\ell_\infty, \ell_2, \ell_1)$ attacks. Our method is no different.
>
>
> 4)    However, Eq. (1) can be treated as mitigating through randomization. Unfortunately, this mechanism has been broken by [R1]. Therefore, the vulnerability of SNAP should be fully considered under various kinds of adversarial methods.
>
> [R1] Obfuscated Gradients Give a False Sense of Security: Circumventing Defenses to Adversarial Examples
>
> We are very well aware of [R1] and the follow-up work by Tramer et al. (2020) (ref. [38] in the main text) on adaptive attacks. In fact, our paper already employs the techniques in [R1] and [38] to confirm that our robustness gains are not due to randomized gradients as described below:
> - Sec. 5.1 describes the detailed evaluation set-up which includes the use of EOT (suggested in [R1]) to eliminate the impact of randomization.
> - The stress tests in Appendix Sec. 1 and Fig. I show the strongest PGD attack includes gradient averaging over 8 noise samples, 10 random restarts, and 500 iterations --> 40,000 backpropagation iterations per image. Such strong attacks had previously eliminated the impact of gradient randomization in both [R1] and Tramer et al., 2020 [38]. However, our robustness gains have sustained.
> - In Appendix Sec. 1, we also evaluate SNAP against gradient-free attacks such as Boundary attack.
> - Based on Reviewer ymM2’s suggestion, we now also evaluate against stronger $ell_2$ AutoAttack to observe TRADES+SNAP achieves 7% and 19% higher $\mathcal{A}^{(\ell_2)}_\text{adv}$ compared to TRADES for $\epsilon = 0.5$ and $\epsilon = 1.0$, respectively. EOT is included in the AutoAttack as well.
>
> All the above confirms that SNAP is in fact **not** broken by [R1].
>
> We hope that you will consider increasing your rating in light of our responses. We are happy to continue this conversation during the discussion period.

---

> ### Author Response · Authors · 2021-08-31
> **Query**
>
> Dear Reviewer d8RZ: We appreciate your review and hope that you have found our rebuttal satisfactory. If not, please do let us know of any additional concerns you may have and we will be happy to answer those. Please note that our work has been thoroughly vetted in the past and our results can be replicated by the readers using our code that will be made available after the paper decision. You have probably noticed that our work is the first to robustify ResNet-50 and ResNet-101 on ImageNet against union of norm-bounded perturbations, thanks to its low complexity. We are glad to answer any additional questions/concerns you may have including providing additional simulations results provided it is feasible to do so in the time period available.

---

> ### Author Response · Authors · 2021-09-03
> **Further clarification**
>
> In addition to our earlier response, we would like to provide a further clarification about point 3) above (repeated below), which seems to be the reviewer’s main concern.
>
> 3(a) **Unfortunately, this mechanism has been broken by [R1].**
>
> [R1] Obfuscated Gradients Give a False Sense of Security: Circumventing Defenses to Adversarial Examples, 2018
>
> SNAP is in fact **not** broken by [R1] because we are *already* employing the EOT (suggested in [R1]) to eliminate the impact of randomization during our PGD attack. Our setup follows the PGD formulation by (Salman et al., NeurIPS 2019) for **smoothed** deep nets and it is known to be the strongest PGD attack against the **smoothed** deep nets, such as SNAPnets.
>
> Furthermore, TRADES+SNAP outperforms TRADES *even after removing the noise entirely at the test time* (see point (1) in our response to the Reviewer ymM2 for the detailed discussion). This clearly shows that SNAP did change the decision boundary appropriately during training and that its benefits are **not** solely due to randomization. Kindly find these new experimental results below.
>
> The Table below shows $\mathcal{A}^{(\ell_2)}_\text{adv}$ (%) for TRADES and TRADES+SNAP (without noise during PGD and test) against the PGD-100 attack with 10 random restarts.
>
> |$\epsilon_{\ell_2}$| TRADES |  TRADES+SNAP (no noise during PGD and test) |
> |:---:|:---:|:---:|
> | 0.5  | 59.6  | 61.9  |
> | 1.0  | 30.4  |  37.5  |
>
> The Table below shows $\mathcal{A}^{(\ell_1)}_\text{adv} $ (%) for TRADES, TRADES+SNAP, and TRADES+SNAP (without noise during test) against the PGD-100 attack with 10 random restarts.
>
> |$\epsilon_{\ell_1}$| TRADES |  TRADES+SNAP (no noise during PGD and test) |
> |:---:|:---:|:---:|
> | 12 | 19.8  |  39.2 |
> | 16  | 10.7  | 28.6  |
>
> TRADES+SNAP (without test noise) achieves significantly higher robust accuracies compared to TRADES. For example, with  $\epsilon_{\ell_1}$=12, TRADES+SNAP achieves 19% higher $A^{(\ell_1)}_\text{adv}$ compared to TRADES *even after removing the noise*.
>
> 3(b) **Therefore, the vulnerability of SNAP should be fully considered under various kinds of adversarial methods.**
>
> Indeed we do evaluate against the additional adversarial methods as a part of our response. Following Reviewer ymM2’s suggestion, we have evaluated the SNAP models against $\ell_2$ and $\ell_\infty$ bounded AutoAttacks, which are the new benchmark attacks and are known to detect/eliminate the impact of gradient randomization.
>
> Our robust accuracies are sustained even against the AutoAttacks and we see the same trends as that of our PGD evaluations.
>
> For example, the table below presents $\mathcal{A}^{(\ell_2)}_\text{adv}$ (%) against AutoAttack (with N=8 EOT) for TRADES and TRADES+SNAP
>
> |$\epsilon_{\ell_2}$| TRADES | TRADES+SNAP|
> |:---:|:---:|:---:|
> | 0.1  | 79.1  | 76.9  |
> | 0.25  | 72.0  |  74.2 |
> |  0.5 | 57.9  | 65.8  |
> | 0.75  | 43.3  | 56.5  |
> |  1.0 |  27.2  |  46.9 |
> | 1.25  |  14.7 |  35.9 |
> |  1.5 | 7.1  | 27.0  |
> | 1.75  | 3.1  |  17.7 |
> | 2.0  | 1.4  | 11.2   |
> | 2.25  |  0.6 |  5.7 |
> |  2.5 | 0.3  | 2.9  |
>
> Similarly, the table below presents $\mathcal{A}^{(\ell_\infty)}_\text{adv}$ (%) against AutoAttack (with N=8 EOT) for TRADES and TRADES+SNAP
>
> | $\epsilon_{\ell_\infty}$ | TRADES | TRADES+SNAP|
> |:---:|:---:|:---:|
> | 0.005 | 78.4  | 75.6  |
> | 0.01  | 72.5  |  69.7 |
> |  0.02 | 61.3  |  57.4  |
> | 0.031  | 47.6  | 43.2  |
> |  0.04 |  36.9  |  31.7 |
> | 0.06 |  16.0|  12.1 |
> |  0.08 | 5.9  | 2.7  |
> | 0.1  | 2.2  |  1.1 |
>
> Most notably, TRADES+SNAP achieves 19% higher $\mathcal{A}^{(\ell_2)}_\text{adv}$ compared to TRADES even against AutoAttack when  $\epsilon$=1.0.
>
> Thus, overall, we have evaluated against the strongest PGD (for **smoothed** nets), DDN, Boundary, Square, FAB, and AutoAttacks. We have also swept the number of attack iterations to 500 and checked multiple attack norm values. We have also evaluated the impact removing the noise entirely. Therefore, our evaluations follow the state-of-the-art guidelines for rigorous robustness evaluations.
>
> We would be glad to provide any additional evaluations if the reviewer has any such suggestions.

---

### Official Review · Reviewer_tGS8 · 2021-07-17

**Rating:** 7
**Confidence:** 3

**Summary:**

This paper presents a new adversarial training procedure that enhances existing $L_\infty$ AT methods by adding specific noise to the images, with the goal of improving robustness to multiple types of perturbation ($L_1, L_2, L_\infty$).

The proposed method improves upon existing AT methods in two ways:
1. For the underlying $L_\infty$ procedure, the addition of noise adds considerable robustness to other norms, at the expense of a small reduction in robustness for $L_\infty$.
2. In comparison of other approaches targeting robustness against union of perturbation, the proposed method reduces the runtime by employing a single attack AT, with a relatively cheap noise augmentation. While competing approaches have to compute multiple attacks, the proposed method only computes one attack and only occasionally updates the noise distribution.

The improvement in performance allows the method to be applied to large datasets (ImageNet), which is prohibitive under competing approaches.

**Limitations And Societal Impact:**

The authors have adequately addressed the limitations and societal impact in the final discussion section.

**Main Review:**

Overall, the presentation of the results is well done. The figures and tables are easy to read and well-elaborated with useful visual aids, which I found very helpful in some instances. The experimental setup is well described and the choice and discussion of experiments contribute to a cohesive narrative. While there is a lack of theoretical results and the originality of the method is somewhat reduced, this is compensated by a solid experimental evaluation.

The efficacy of the proposed method is clearly demonstrated with the empirical results.There are relevant comparison with SOTA methods that tackle both reducing complexity of AT and/or improving robustness for multiple attacks. In particular, Table 3 makes a compelling argument for the trade-off between the robustness and the training time. It does a good job of highlighting the benefits of SNAP when the training time is restricted to less than 12 hours.

The flexibility of choosing any base AT procedure and the infrequent updates of the noise distribution made it possible to obtain good robustness results on ImageNet, which is exceedingly costly with other methods.

**Weaknesses:**

In essence, SNAP is a combination of randomized smoothing and adversarial training, i.e. adversarial training with noise-augmented image, where the noise distribution is occasionally updated throughout the training. However, it took me a while to understand this. The clarity of the paper could be improved by describing the similarities (and differences) to a method such as SmoothAdv earlier in the paper. Also, due to the strong parallels between AT+SNAP and SmoothAdv, I expected to see it included in the results of Tables 1 and 2. While there is some discussion in the supplementary material, having this comparison in the mentioned tables could contribute to set SNAP apart from the other randomized smoothing + AT approach.

The spectral subspace analysis, while interesting, was a bit hard to interpret. For example, why the dimensionality of the projection implies that the noise vectors are aligned? Couldn't they still be pointing in different directions even though they lie in a small subspace? Also, in Figure 3a, the dimension seems to be very different for each type of noise. This section brings an interesting discussion about how AT affects the geometry of the perturbations, but it is not easy to interpret and understand the implications of the analysis.



**Time Spent Reviewing:**

5 hours

---

> ### Author Response · Authors · 2021-08-10
> **Authors' response to the Reviewer's comments**
>
> We thank the reviewer for valuable feedback. Here is our response:
>
> 1)    Move the comparison with SmoothAdv to the main text for better clarity:
>
> Thank you for this suggestion. We do agree with your assessment regarding SmoothAdv and SNAP. We will draw these parallels and differences in Section 4. Furthermore, we in fact thoroughly compare SNAP and SmoothAdv in Appendix Figure II and Sec. 2.1. Specifically, we sweep the total noise variance for both SmoothAdv and SNAP to show that SNAP achieves a much better natural accuracy $\mathcal{A}_\text{nat}$ vs robustness  $\mathcal{A}^{(U)}_\text{adv}$ trade-off. As per the reviewer's suggestion, we will include the key points of this comparison in the Main text tables (Table 2 and 3), and discuss them briefly in the Results Section.
>
> 2)    Also, in Figure 3a, the dimension seems to be very different for each type of noise.
>
> Note, the dimensions of the perturbations are identical for each type of noise in Fig. 3a. It seems different because we truncated the **log-scale** y-axis to \[  $ 4 \times 10^{-5}$, $ 4 \times 10^{-3}$ \] in order to highlight the interesting trends. This means the red curve in Fig. 3a continues until $j=3072$.
>
> 3)    why the dimensionality of the projection implies that the noise vectors are aligned? Couldn’t they still be pointing in different directions even though they lie in a small subspace?
>
> You are right. All we are saying is that the three perturbation types are squeezed into a smaller subspace of 250-D after $\ell_\infty$-AT which indicates the plausibility of defending against all three by introducing noise that is shaped to lie predominantly in the dominant subspace of perturbation types. This is the key take away of our Sec. 3 and the corresponding Fig. 3. We will clarify it further in the revision.
>
> We hope that you will consider increasing your rating in light of our responses. We are glad to continue this conversation during the discussion period.

---

> ### Author Response · Authors · 2021-08-31
> **Query**
>
> Dear Reviewer tGS8: We appreciate your reading of our paper and hope that our rebuttal was found satisfactory. Please do let us know if there are any lingering concerns that may be inhibiting an 'acceptance' recommendation to be provided. We can assure you that our work has been thoroughly vetted in the past, and our results can be replicated by the readers. We are glad that you have noticed our ImageNet results as being the first of its kind. We are happy to provide additional clarifications if there are any additional concerns.

---

> ### Comment · Reviewer_tGS8 · 2021-09-02
> **Thank you for the response.**
>
> The response adequately addressed my concerns and clarified some aspects of the spectral analysis. I also reviewed the response to the other reviewers and the new experiments. I believe the clarifications and new results improve the quality of the paper, so I'm increasing my score to recommend acceptance.
>
> I would also recommend a modification to Figure 3, to change the indication of $m_{max}$ and $0.1m_{max}$ to a line (perhaps in different color and/or style) spanning the entire x-axis of the plot, to distinguish it from the arrow segments marking the dimension. This would help to avoid potential confusions. The actual grid in the plot could also be dimmed, to improve readability of the figure.

---

> > ### Author Response · Authors · 2021-09-02
> > **Appreciation**
> >
> > We really appreciate your comments and your recommendation of acceptance. We agree with your suggestions for changes to Figure 3 and will be glad to incorporate those in the final version. Thanks!

---

### Decision · Program_Chairs · 2021-09-27

**Decision:**

Reject

**Comment:**

This is indeed a borderline paper. Based on the majority of opinions from reviewers and the suggested accept rate of program committee, a current rejection decision is made.